# GRAPH METANETWORKS FOR PROCESSING DIVERSE NEURAL ARCHITECTURES

**Derek Lim** *
MIT CSAIL
dereklim@mit.edu

**Haggai Maron**
Technion / NVIDIA
hmaron@nvidia.com

**Marc T. Law**
NVIDIA
marcl@nvidia.com

**Jonathan Lorraine**
NVIDIA
jlorraine@nvidia.com

**James Lucas**
NVIDIA
jalucas@nvidia.com

## ABSTRACT

Neural networks efficiently encode learned information within their parameters. Consequently, many tasks can be unified by treating neural networks themselves as input data. When doing so, recent studies demonstrated the importance of accounting for the symmetries and geometry of parameter spaces. However, those works developed architectures tailored to specific networks such as MLPs and CNNs without normalization layers, and generalizing such architectures to other types of networks can be challenging. In this work, we overcome these challenges by building new metanetworks — neural networks that take weights from other neural networks as input. Put simply, we carefully build graphs representing the input neural networks and process the graphs using graph neural networks. Our approach, Graph Metanetworks (GMNs), generalizes to neural architectures where competing methods struggle, such as multi-head attention layers, normalization layers, convolutional layers, ResNet blocks, and group-equivariant linear layers. We prove that GMNs are expressive and equivariant to parameter permutation symmetries that leave the input neural network functions unchanged. We validate the effectiveness of our method on several metanetwork tasks over diverse neural network architectures.

## 1  INTRODUCTION

Neural networks are well-established for predicting, generating, and transforming data. A newer paradigm is to treat the parameters of neural networks themselves as data. This insight inspired researchers to suggest neural architectures that can predict properties of trained neural networks (Eilertsen et al., 2020), generate new networks (Erkoç et al., 2023), optimize networks (Metz et al., 2022), or otherwise transform them (Navon et al., 2023; Zhou et al., 2023a). We refer to these neural networks that process other neural networks as *metanetworks*, or *metanets* for short.

Metanets enable new applications, but designing them is nontrivial. A common approach is to flatten the network parameters into a vector representation, neglecting the input network structure. More generally, a prominent challenge in metanet design is that the space of neural network parameters exhibits symmetries. For example, permuting the neurons in the hidden layers of a Multilayer Perceptron (MLP) leaves the network output unchanged (Hecht-Nielsen, 1990). Ignoring these symmetries greatly degrades the metanet performance (Peebles et al., 2022; Navon et al., 2023). Instead, equivariant metanets respect these symmetries, so that if the input network is permuted then the metanet output is permuted in the same way.

Recently, several works have proposed equivariant metanets that have shown significantly improved performance (Navon et al., 2023; Zhou et al., 2023a;b). However, these networks typically require highly specialized, hand-designed layers that can be difficult to devise. A careful analysis of its symmetries is necessary for any input architecture, followed by the design of corresponding equivariant metanet layers. Generalizing this procedure to more complicated network architectures is

---

*Work done while interning at NVIDIA.

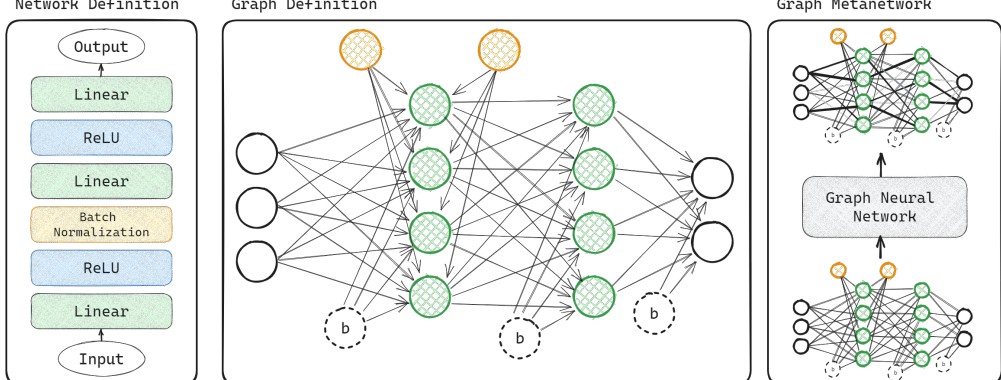

Figure 1: **Overview of Graph Metanetworks (GMNs)** Our method converts neural network architectures into a parameter graph where edges correspond to network parameters. The bias ($b$) and batch-normalization parameters are incorporated via additional nodes with edges to the relevant layer's neurons. The graph is processed by a graph neural network operating on edge attributes. Fixed-length (invariant) predictions can be extracted by pooling the output graph features.

time-consuming and nontrivial, so existing methods can only process simple input networks with linear and convolutional layers, and cannot process standard modules such as normalization layers or residual connections — let alone more complicated modules such as attention blocks. Moreover, these architectures cannot directly process input neural networks with varying architectures, such as those with different numbers of layers or hidden units.

This work offers a simple and elegant solution to metanet design that respects neural network parameter symmetries. As in the concurrent work of Zhang et al. (2023), our technique's crux is representing an input neural network as a graph (see Figure 1). We show how to efficiently transform a neural network into a graph such that standard techniques for learning on graphs – e.g., Message Passing Neural Networks Gilmer et al. (2017); Battaglia et al. (2018) or Graph Transformers (Rampášek et al., 2022) – will be equivariant to the parameter symmetries. One of our key contributions is in developing a compact *parameter graph* representation, which in contrast to established computation graphs allows us to handle parameter-sharing layers like convolutions and attention layers without scaling with the activation count. While past work is typically restricted to processing MLPs and simple Convolutional Neural Networks (CNNs) (LeCun et al., 1989), we validate experimentally that our graph metanets (GMNs) generalize to more complicated networks such as Transformers (Vaswani et al., 2017), residual networks (He et al., 2016), normalization layers (Ioffe & Szegedy, 2015; Ba et al., 2016; Wu & He, 2018), general group-equivariant architectures (Ravanbakhsh et al., 2017) like Deep Sets (Zaheer et al., 2017), and more.

We prove theoretically that our metanets are equivariant to permutation symmetries in the input network, which we formulate via neural graph automorphisms (Section 2.2). This generalizes the hidden neuron permutations in MLPs and channel permutations of CNNs covered in prior work to arbitrary feedforward neural architectures. We further prove that our metanets operating on computation graphs are at least as expressive as prior methods — meaning they can approximate them to arbitrary accuracy — and can express the forward pass of any input feedforward neural network, generalizing a result of Navon et al. (2023) (Section 3).

Empirical evaluations show that our approach solves a variety of metanet tasks with diverse neural architectures. As part of this effort, we trained new datasets of diverse image classifiers, including 2D CNNs, 1D CNNs, DeepSets, ResNets, and Vision Transformers. Our method is easier to implement than past equivariant metanets while being at least as expressive, and it is applicable to more general input architectures. Crucially, our GMNs achieve strong quantitative performance across all tasks we explored.

## 2 GRAPH AUTOMORPHISM-BASED METANETS

We first explain how neural networks can be encoded as Directed Acyclic Graphs (DAGs). There are many choices in representing neural networks as DAGs, perhaps the most common being a

computation graph (see Appendix C). This work introduces a more compact representation, referred to as *parameter graphs*.

We then introduce one of the paper's main concepts — *Neural DAG Automorphisms*. This concept generalizes previously studied symmetry groups for MLPs and CNNs to arbitrary feedforward architectures represented as DAGs. To conclude this section, we describe our GNN-based metanet that operates over these graphs and is equivariant to Neural DAG Automorphisms. A glossary of our notation is provided in Appendix Table 4.

**Motivation.** Certain permutations of parameters in neural networks do not change the function they parameterize. For example, consider a simple MLP defined such that $f_{\boldsymbol{\theta}}(\boldsymbol{x}) := \boldsymbol{W}_2 \sigma(\boldsymbol{W}_1 \boldsymbol{x})$ with one hidden layer, where $\boldsymbol{\theta} := (\boldsymbol{W}_2, \boldsymbol{W}_1)$ are the parameters of the network, and $\sigma$ is a nonlinear element-wise activation function. For any permutation matrix $\boldsymbol{P}$, if we define $\tilde{\boldsymbol{\theta}} := (\boldsymbol{W}_2 \boldsymbol{P}^\top, \boldsymbol{P} \boldsymbol{W}_1)$, then for all inputs $\boldsymbol{x}$, we have $f_{\boldsymbol{\theta}}(\boldsymbol{x}) = f_{\tilde{\boldsymbol{\theta}}}(\boldsymbol{x})$. This $\boldsymbol{P}$ corresponds to a permutation of the order of the hidden neurons, which is well-known not to affect the network function (Hecht-Nielsen, 1990). Likewise, permuting the hidden channels of a CNN does not affect the network function (Navon et al., 2023; Entezari et al., 2022; Ainsworth et al., 2023).

While these permutation symmetries for MLPs and simple CNNs are easy to determine by manual inspection, it is more difficult to determine the symmetries of general architectures. For example, simple residual connections introduce additional neuron dependencies across layers. Instead of manual inspection, we show that graph automorphisms (i.e. graph isomorphisms from a graph to itself) on DAGs representing feedforward networks correspond to permutation parameter symmetries. From this observation, it can be shown that GNNs acting on these DAGs are equivariant to their permutation symmetries.

**Overview of our approach.** See Figure 1 for an illustration. Given a general input feedforward neural network, we first encode it as a graph in which each parameter is associated with an edge and then process this graph with a GNN. The GNN outputs a single fixed-length vector or predictions for each node or edge depending on the learning task. For instance, one graph-level task is to predict the scalar accuracy of an input neural network on some task. An edge-level task is to predict new weights for an input neural network to change its functionality somehow.

We now discuss the graph construction, the symmetries of these graphs, and the GNN we use.

## 2.1 GRAPH CONSTRUCTION FOR GENERAL FEEDFORWARD ARCHITECTURES

### 2.1.1 COMPUTATION GRAPHS

Every feedforward neural network defines a computation graph as a DAG (Zhang et al., 2023), where nodes are neurons and edges hold neural network parameter weight values (see Fig. 1 and Fig. 2). Thus, this gives a method to construct a weighted graph. However, the computation graph approach has some downsides. For one, it may be expensive due to weight-sharing: for instance, a 1-input-and-output-channel 2D-convolution layer with a kernel size of 2 has 4 parameters, but the 4 parameters are used in the computation of many activations (e.g. 1024 activations for a $32 \times 32$ input grid). Further, we may want to add input node and edge features – such as layer number – to help performance and expressive power[1]. Figure 2 illustrates an example of a (small) computation

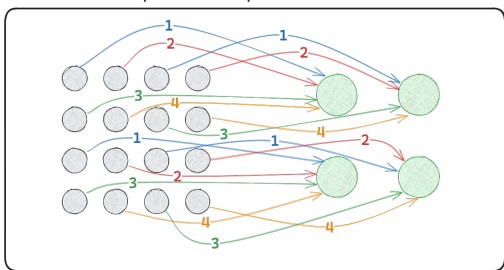

Figure 2: An example computation graph for a network with a single convolutional layer. The layer has a $2 \times 2$ filter kernel, a single input and output channel, and applies the filter with a stride of 2. Even in this small case of a $4 \times 4$ input image, the graph has 16 edges for only 4 parameters.

graph for convolutions (for visual clarity, we exclude bias terms). More details, including the exact formal correspondence between feedforward neural network functions and computation graphs, are given in Appendix C.1.

---

[1] In Section 3, our proofs rely on these features to show graph metanets can express existing metanets.

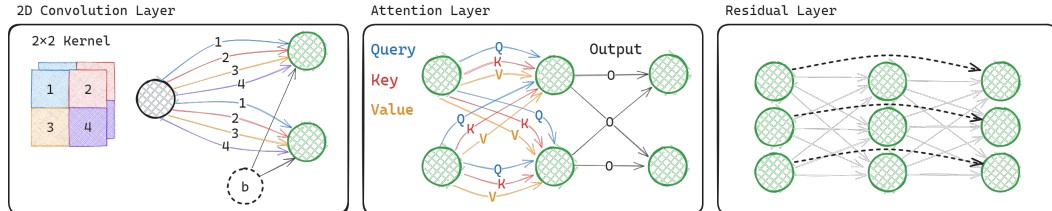

Figure 3: Parameter subgraph constructions for assorted layers that we implemented in our empirical evaluation. Their descriptions are given in Section 2.1.2. Further details are discussed in Appendix B.

### 2.1.2 PARAMETER GRAPHS

To deal with the challenges of computation graphs, we propose alternate neural network graph constructions — some examples of which are shown in Figure 3 — that are (a) efficient and (b) allow expressive metanets. We call these graphs *parameter graphs* because we design the graphs so that each parameter is associated with a single edge of the graph (whereas a parameter may be associated to many edges in a computation graph).

We design modular subgraphs that can be created for each layer and then stitched together. Our goal is to design parameter graphs with at most one edge for each parameter in the neural network. Additionally, they should capture the correct set of parameter permutation symmetries. Full details are in Appendix B, but we discuss the major points behind the design of a selection of parameter graphs here.

**Linear layers.** Figure 1 depicts three linear layers in sequence. Each linear layer's parameter subgraph coincides with its computation graph, but even so, there are important design choices to be made. Bias parameters could be encoded as node features as done by Zhang et al. (2023) or as self-loop edges on each neuron. Instead, we include a bias node for each layer and encode the bias parameters as edges from that node to the corresponding neurons in the layer.

The bias node approach is preferable because the self-loop or node feature approaches to encoding biases can hinder the expressive power of the metanet. In particular, the results of Navon et al. (2023) and Zhou et al. (2023a) show that the permutation equivariant linear metanet maps for MLP inputs include message-passing-like operations where the representation of a bias parameter is updated via the representations of other bias parameters in the same layer. Using a message passing GNN on a graph with bias nodes allows us to naturally express these operations in a single graph metanet layer, as explained in Appendix D.1.3.

**Convolution layers.** Convolutions and other group equivariant linear layers leverage parameter sharing, where the same parameter is used to compute many activations (Ravanbakhsh et al., 2017). Therefore, representing convolutions as a computation graph introduces scaling issues and binds the network graph to a choice of input size. To avoid this, we develop a succinct and expressive parameter graph representation of convolutions. This is depicted in the left of Figure 3 for a convolution layer with a $2 \times 2$ filter, one input channel, and two output channels. Note that we have two output channels here, unlike the computation graph in Figure 2, where there is only one.

Our subgraph construction allocates a node for each input and output channel. We then have parallel edges between each input and output node for each spatial location in the filter kernel — making this a multigraph. Bias nodes are added as in the linear layers. This subgraph contains exactly one edge for each parameter in the convolution layer while capturing the parameter permutation symmetries as graph automorphisms. The spatial position of each weight within the kernel is included by a positional encoding in the corresponding edge feature.

In Section 4.1, we use our graph metanets to process 1D and 2D convolutional networks, as well as DeepSets networks that consist of permutation equivariant linear layers (Zaheer et al., 2017).

**Multi-head attention layers.** Attention layers (Vaswani et al., 2017) also exhibit parameter sharing across sequence length. As with convolutions, we design an efficient subgraph representation where each parameter appears as a single edge. There is one node for each feature dimension of the input,

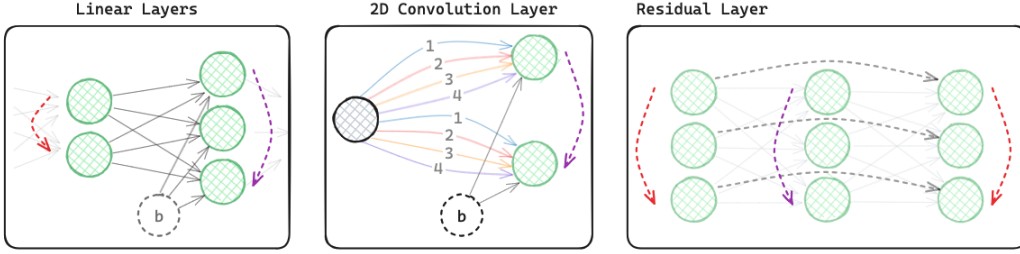

Figure 4: Examples of neural DAG automorphisms for linear layers, convolutional layers, and residual layers. Possible node permutations are illustrated using red and purple dashed arrows, the same color represents an identical transformation.

vectors used in attention computation, and output. There are two sets of edges: a set that corresponds to the query, key, and value maps, and a second set corresponding to the final output mapping.

In the middle of Figure 3 we show a single-headed self attention layer, with bias nodes excluded for visual clarity. Generalizing this to multi-head attention simply requires adding additional node features to the middle layer that indicate which head each node belongs to.

**Residual layers.** A residual connection does not introduce any additional parameters, but it does affect the permutation symmetries in the parameter space of the network. Therefore, it is crucial to represent residual connections as additional parameter-free edges within the parameter graph. The top-right of Figure 3 shows a residual connection bypassing a linear layer. The edges are drawn as dashed lines to emphasize that there are no associated parameters. As is natural in the computation graph, we fix the weight of the residual edge to be 1.

## 2.2 NEURAL DAG AUTOMORPHISMS

The prior section describes how to represent (feedforward) neural networks as DAGs. A natural question from an equivariant learning perspective is: what are the symmetries of this DAG representation? Specifically, we consider graph automorphisms, which are structure-preserving transformations of a graph unto itself. A neural DAG automorphism of a DAG $(V, E)$ associated with a neural network is a permutation of nodes $\phi : V \to V$ that preserves adjacency, preserves types of nodes (e.g. $\phi$ cannot map hidden neurons to input neurons), and preserves weight-sharing constraints (i.e. tied weights must still be tied after permuting the endpoints with the automorphism); see Appendix C.2 for more details. Every automorphism $\phi$ also induces a permutation of edges $\phi^{\boldsymbol{e}} : E \to E$, where edge $(i, j)$ is mapped to $\phi^{\boldsymbol{e}}((i, j)) = (\phi(i), \phi(j))$.

Intuitively, a neural DAG automorphism represents a permutation of the neural network parameters via the induced edge permutation, $\phi^e$. We write $\Phi(\boldsymbol{\theta})$ to represent this permutation on the parameters themselves, meaning $\Phi(\boldsymbol{\theta})_{(\phi(i),\phi(j))} = \boldsymbol{\theta}_{(i,j)}$. Hidden node permutations in MLPs and hidden channel permutations in CNNs are special cases of neural DAG automorphisms, which we explain in Appendix C.3. To formalize these notions, we show that every neural DAG automorphism $\phi$ of a computation graph is a permutation parameter symmetry, in the sense that the induced parameter permutation $\Phi$ does not change the neural network function. Figure 4 illustrates several neural DAG automorphisms.

**Proposition 1.** *For any neural DAG automorphism $\phi$ of a computation graph, the neural network function is left unchanged:* $\forall \boldsymbol{x} \in \mathcal{X}, f_{\boldsymbol{\theta}}(\boldsymbol{x}) = f_{\Phi(\boldsymbol{\theta})}(\boldsymbol{x})$.

A proof is given in Appendix C.4. Recall that our goal is to design metanets equivariant to parameter permutation symmetries. Proposition 1 shows to achieve this it is necessary to design metanets that are equivariant to neural DAG automorphisms. Graph metanets achieve this exactly since GNNs are equivariant to permutation symmetries of graphs (Maron et al., 2019).

**Proposition 2.** *Graph metanets are equivariant to parameter permutations induced by neural DAG automorphisms.*

These results formally justify using graph metanets on computation graphs for equivariance to parameter permutation symmetries. Now, noting the parameters are stored as edge features in our computation and parameter graphs, we design graph neural networks that operate on these DAGs.

## 2.3 FORMULATING METANETS AS GNNS

After constructing the input graphs, we use a GNN as our metanet to learn representations and perform downstream tasks. We desire GNNs that learn edge representations since the input neural network parameters are placed on the edges of the constructed graph. While countless GNN variants have been developed in the last several years (Hamilton, 2020), most learn node representations.

For simplicity, we mostly loosely follow the general framework of Battaglia et al. (2018), which defines general message-passing GNNs that update node, edge, and global features. For a graph, let $\boldsymbol{v}_i \in \mathbb{R}^{d_x}$ be the feature of node $i$, $\boldsymbol{e}_{(i,j)} \in \mathbb{R}^{d_e}$ a feature of the directed edge $(i,j)$, $\boldsymbol{u} \in \mathbb{R}^{d_u}$ be a global feature associated to the entire graph, and let $E$ be the set of edges in the graph. The directed edge $(i,j)$ represents an edge starting from $j$ and ending at $i$. We allow multigraphs, where there can be several edges (and hence several edge features) between a pair of nodes $(i,j)$; thus, we let $E_{(i,j)}$ denote the set of edge features associated with $(i,j)$. Then, a general GNN layer updating these features can be written as:

$$\boldsymbol{v}_i \leftarrow \text{MLP}_2^{\boldsymbol{v}} \left( \boldsymbol{v}_i, \sum_{j, \boldsymbol{e}_{(i,j)} \in E_{(i,j)}} \text{MLP}_1^{\boldsymbol{v}}(\boldsymbol{v}_i, \boldsymbol{v}_j, \boldsymbol{e}_{(i,j)}, \boldsymbol{u}), \boldsymbol{u} \right) \tag{1}$$

$$\boldsymbol{e}_{(i,j)} \leftarrow \text{MLP}^{\boldsymbol{e}}(\boldsymbol{v}_i, \boldsymbol{v}_j, \boldsymbol{e}_{(i,j)}, \boldsymbol{u}) \tag{2}$$

$$\boldsymbol{u} \leftarrow \text{MLP}^{\boldsymbol{u}} \left( \sum_i \boldsymbol{v}_i, \sum_{\boldsymbol{e} \in E} \boldsymbol{e}, \boldsymbol{u} \right) \tag{3}$$

Intuitively, node features are updated by message passing along neighbors (Equation 1), the features of adjacent nodes update the features of the edges connecting them (Equation 2), and the global feature $\boldsymbol{u}$ is updated with aggregations of all features (Equation 3). While our graphs are DAGs, we are free to use undirected edges by ensuring that $(i,j) \in E$ implies $(j,i) \in E$ with $\boldsymbol{e}_{(i,j)} = \boldsymbol{e}_{(j,i)}$. We often choose to allow message passing between layers in both directions.

For parameter-level metanet tasks with per-parameter predictions, we let the prediction be the final feature $\boldsymbol{e}_{(i,j)}$ of the parameter's corresponding edge. We pool edge features for network-level metanet tasks where a fixed-length vector is the final prediction for each graph. We can pool node features but found pooling edge features sufficient. We do not use global features for empirical results, but we find them crucial for the expressive power results of Proposition 3.

## 3 EXPRESSIVE POWER OF GRAPH METANETS (GMNS)

Ideally, one does not sacrifice expressive power when restricting a neural architecture to satisfy group equivariance constraints. We want our metanet architecture to be powerful enough to learn useful functions of network parameters. To this end, we first show that GMNs can express two existing equivariant metanets on MLP inputs. Consequently, GMNs are at least as expressive as these approaches.

**Proposition 3.** *On MLP inputs (where parameter graphs and computation graphs coincide), graph metanets can express StatNN (Unterthiner et al., 2020) and NP-NFN (Zhou et al., 2023a).*

The proof is given in Appendix D.1. StatNN is based on permutation-invariant statistics of each weight or bias, which graph metanetworks can easily compute. The linear layers of NP-NFN consist of local message-passing-like operations and global operations that the global feature can capture.

Second, we show that GMNs can simulate the forward pass of an input neural network represented by the DAG of its computation graph (Appendix C.1). This substantially generalizes a result of Navon et al. (2023), who show that their DWSNets can simulate the forward pass of MLPs.

**Proposition 4.** *On computation graph inputs, graph metanets can express the forward pass of any input feedforward neural network as defined in Section C.1.*

Table 1: Results for predicting the test accuracy of input neural networks trained on CIFAR-10. The top results use a training set of 15 000 uniformly selected input networks, the middle results use 10% of this training set, and the bottom results only train on input networks of low hidden dimension (while testing on networks with strictly higher hidden dimension). Our method performs best in all setups, with increasing benefits in the low-data and OOD regimes.

| | Metanet | Varying CNNs | | Diverse Architectures | |
| | | $R^2$ | $\tau$ | $R^2$ | $\tau$ |
|---|---|---|---|---|---|
| 50% | DeepSets (Zaheer et al., 2017) | $.778{\pm}.002$ | $.697{\pm}.002$ | $.562{\pm}.020$ | $.559{\pm}.011$ |
| | DMC (Eilertsen et al., 2020) | $.948{\pm}.009$ | $.876{\pm}.003$ | $.957{\pm}.009$ | $.883{\pm}.007$ |
| | GMN (Ours) | $\mathbf{.978}{\pm}.002$ | $\mathbf{.915}{\pm}.006$ | $\mathbf{.975}{\pm}.002$ | $\mathbf{.908}{\pm}.004$ |
| 5% | DeepSets (Zaheer et al., 2017) | $.692{\pm}.006$ | $.648{\pm}.002$ | $.126{\pm}.015$ | $.290{\pm}.010$ |
| | DMC (Eilertsen et al., 2020) | $.816{\pm}.038$ | $.762{\pm}.014$ | $.810{\pm}.046$ | $.758{\pm}.013$ |
| | GMN (Ours) | $\mathbf{.876}{\pm}.010$ | $\mathbf{.797}{\pm}.005$ | $\mathbf{.918}{\pm}.002$ | $\mathbf{.828}{\pm}.005$ |
| OOD | DeepSets (Zaheer et al., 2017) | $.741{\pm}.015$ | $.683{\pm}.005$ | $.128{\pm}.071$ | $.380{\pm}.014$ |
| | DMC (Eilertsen et al., 2020) | $.387{\pm}.229$ | $.760{\pm}.024$ | $-.134{\pm}.147$ | $.566{\pm}.055$ |
| | GMN (Ours) | $\mathbf{.891}{\pm}.037$ | $\mathbf{.870}{\pm}.010$ | $\mathbf{.768}{\pm}.063$ | $\mathbf{.780}{\pm}.030$ |

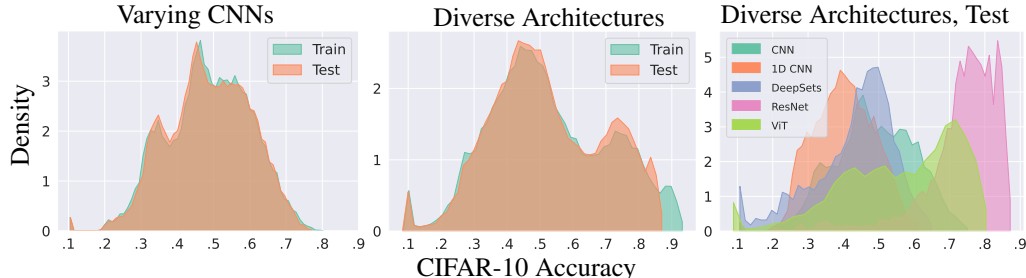

Figure 5: Histograms of CIFAR-10 accuracies for our Varying CNNs and Diverse Architectures datasets. Left and middle show train and test accuracy for the two datasets. Right shows test accuracy of Diverse Architectures split by model type.

The above result applies only to GMNs operating on computation graphs. This makes it directly applicable to the parameter graphs for MLPs as they coincide with the computation graph. However, we expect that a similar result is possible for more general parameter graphs. We leave formal proof of this to future work.

# 4 EXPERIMENTS

## 4.1 PREDICTING ACCURACY FOR VARYING ARCHITECTURES

**Task.** As in prior works (Unterthiner et al., 2020; Zhou et al., 2023a), we train metanets to predict the test accuracy of input neural networks. We consider image classification neural networks trained on the CIFAR-10 dataset (Krizhevsky, 2009), and the metanet task is to take the parameters of an input network and predict the network's image classification accuracy on the CIFAR-10 test set.

**Datasets.** To demonstrate the flexibility of our graph metanets, we train image classifiers that significantly vary in size and architecture. For our "Varying CNNs" dataset, we train about 30 000 basic 2D CNNs varying in common architectural design choices like hidden dimension, number of convolution layers, number of fully connected layers, and type of normalization layers (BatchNorm (Ioffe & Szegedy, 2015) or GroupNorm (Wu & He, 2018)).

For our "Diverse Architectures" dataset, we also train four other diverse image classifiers to test our model's generalization with multiple architectures: (i) basic 1D CNNs treating images as raster ordered sequences of pixels, (ii) DeepSets treating images as pixel sets, which maintains pixel position information with positional encodings (Zaheer et al., 2017), (iii) ResNets (He et al., 2016), and (iv) Vision Transformers (Dosovitskiy et al., 2021). This dataset also totals about 30 000 networks. The Vision Transformers' patch embedding module can be viewed as a convolution, which is how we encode this module as a graph (the Transformer layer graph encodings are described in Figure 7).

Figure 5 shows these trained networks span a wide range of accuracies, between 10% and 77.5% for Varying CNNs and between 7.5% and 87.8% for Diverse Architectures. Also, the number of parameters in these networks ranges from 970 to 21 165 in Varying CNNs and from 970 to 87 658 for Diverse Architectures. These networks are significantly more diverse and achieve higher accuracy than the dataset of Unterthiner et al. (2020), who train small CNNs of a fixed architecture that obtain at most 56% test accuracy on CIFAR-10. Also, our Diverse Architectures dataset contains many more network types and modules than the datasets of Eilertsen et al. (2020) and Schürholt et al. (2022), which are limited to CNNs.

**Metanetworks.** For our graph metanet, we consider a simple message passing GNN as in Section 2.3 that does not use a global graph feature. To obtain an invariant prediction, we mean-pool over edge representations.

We cannot apply competing permutation equivariant methods like DWSNet (Navon et al., 2023) or NFN (Zhou et al., 2023a), because they cannot process normalization layers, input neural networks of different sizes, or modules like self-attention. Instead, as a baseline, we consider the Deep Meta Classifier (DMC) from Eilertsen et al. (2020), which vectorizes an input network's parameters and processes it with a 1D CNN, allowing the use of differently sized networks. We also consider a baseline that treats the parameters as a set and applies a DeepSets network (Zaheer et al., 2017) to output a scalar prediction. Note that the DeepSets baseline is invariant to permutation parameter symmetries, but it is also invariant to permutations that do not correspond to parameter symmetries (which significantly outnumber permutation parameter symmetries), so it has low expressive power.

We evaluate our method and the two baselines across six different data settings. Using both the Varying CNN dataset and the Diverse Architectures dataset. We explore training on about half of the input networks, training on only 10% of this previous split, and an out-of-distribution (OOD) generalization setting where we train on a reduced set of architectures that have smaller hidden dimension than the held-out architectures.

**Results.** See Table 1 for quantitative results and Figure 8 in the Appendix for scatterplots in the OOD setting on Diverse Architectures. We report the R-Squared value and the Kendall $\tau$ coefficient of the predicted generalization against the true generalization. Our GMNs outperform both baselines in predicting accuracy for input networks across all six data settings. When we restrict to 10% of the full training set size, we see that GMNs generalize substantially better than the baselines. This performance gap is maintained in the more challenging OOD generalization setting, where the non-equivariant DMC performs very poorly in $R^2$. The improved GMN performance could be from high expressive power (which the DeepSets baseline lacks), with better generalization due to equivariance to parameter permutation symmetries (which DMC lacks).

## 4.2 EDITING 2D INRS

Table 2: Test MSE (lower is better) for editing 2D INRs, following the methodology of (Zhou et al., 2023a). Results of baselines are from (Zhou et al., 2023a;b).

| Metanetwork | Contrast | Dilate |
|---|---|---|
| MLP | .031 | .306 |
| MLP-Aug | .029 | .307 |
| NFN-PT | .029 | .197 |
| NFN-HNP | .0204 $\pm$.0000 | .0706 $\pm$.0005 |
| NFN-NP | .0203 $\pm$.0000 | .0693 $\pm$.0009 |
| NFT | .0200 $\pm$.0002 | **.0510** $\pm$.0004 |
| GMN (ours) | **.0197** $\pm$.0000 | .0603 $\pm$.0010 |

Table 3: Results for self-supervised learning of neural net representations, in test MSE of a linear regressor on the learned representations. Numbers besides GMN from Navon et al. (2023).

| Metanetwork | Test MSE |
|---|---|
| MLP | 7.39 $\pm$.19 |
| MLP + Perm. aug | 5.65 $\pm$.01 |
| MLP + Alignment | 4.47 $\pm$.15 |
| INR2Vec (Arch.) | 3.86 $\pm$.32 |
| Transformer | 5.11 $\pm$.12 |
| DWSNets | 1.39 $\pm$.06 |
| GMN (ours) | **1.13** $\pm$.08 |

Next, we empirically test the ability of GMNs to process simple MLP inputs to compare against less-flexible permutation equivariant metanets such as NFN (Zhou et al., 2023a) and NFT (Zhou et al., 2023b). For this, we train metanetworks on the 2D INR editing tasks of (Zhou et al., 2023a), where the inputs are weights of an INR representing an image, and the outputs are weights of an INR representing the image with some transformation applied to it.

Table 2 shows our results. We see that our GMNs outperform most metanetworks on these simple input networks. In particular, GMN outperforms all methods on the Contrast task, and is only beat by NFT on the Dilate task.

## 4.3 Self-Supervised Learning with INRs

We also compare GMNs against another less-flexible permutation equivariant metanet, DWS-Nets (Navon et al., 2023), in the self-supervised learning experiments of Navon et al. (2023). Here, the input data are MLPs fit to sinusoidal functions of the form $x \mapsto a\sin(bx)$, where $a, b \in \mathbb{R}$ are varying parameters. The goal is to learn a metanet encoder that gives strong representations of input networks, using a contrastive-learning framework similar to SimCLR (Chen et al., 2020). Our GMNs learn edge representations, which are then mean-pooled to get a vector representation of each input network. The downstream task for evaluating these representations is fitting a linear model on the metanet representations to predict the coefficients $a$ and $b$ of the sinusoid that the input MLP represents.

Table 3 shows the results. GMNs outperform all baselines in this task. Thus, even in the most advantageous setting for competitors, which they are restricted to, GMNs are empirically successful at metanet tasks.

## 5 Conclusion

In this work, we proposed Graph Metanetworks, an approach to processing neural networks with theoretical and empirical benefits. Theoretically, our approach satisfies permutation parameter symmetry equivariance while having provably high expressive power. Empirically, we can process diverse neural architectures, including layers that appear in state-of-the-art models, and we outperform existing metanetwork baselines across all tasks that we evaluated.

**Limitations** We make substantial progress towards improving the scalability of GMNs by introducing parameter graphs. However, large neural networks can have billions of parameters and processing them may be more difficult. We believe that standard scalable GNN methods can be used to scale to the billion parameter regime (as existing GNNs are capable of processing graphs with billions of edges (Hu et al., 2021) using modest computing resources), but we have not yet tried to use Graph Metanets at such a scale. Additionally, we argue that parameter graphs are easier to design than specialized architectures of prior work, but we do not give formal constraints on their design. Further work investigating parameter graphs is promising; for instance, our theory of neural DAG automorphisms depends on the more expensive computation graphs, but it could possibly be extended to parameter graphs. Moreover, our approach only accounts for permutation-based parameter symmetries and does not account for e.g. symmetries induced by scaling weights in ReLU networks (Dinh et al., 2017; Godfrey et al., 2022).

**Future work** Our graph-based approach to metanets is promising for future development. This paper mostly uses a basic message-passing GNN architecture, which can be further improved using the many GNN improvements in the literature. Furthermore, our theoretical developments largely apply to computation graphs and we expect future work can extend these results to the more practical parameter graphs. Since graph metanets can process modern neural network layers like self-attention and spatial INR feature grids, they can be used to process and analyze state-of-the-art neural networks. Further, future work could try applying graph metanets to difficult yet impactful metanets tasks such as pruning, learned optimization, and finetuning pretrained models.

### Acknowledgments

We thank Matan Atzmon, Jiahui Huang, Karsten Kreis, Francis Williams, and Xiaohui Zeng for helpful comments. We would also like to thank Jun Gao, Or Perel, and Frank Shen for helpful input on some of our early INR experiments. DL is funded by an NSF Graduate Fellowship. HM is the Robert J. Shillman Fellow, and is supported by the Israel Science Foundation through a personal grant (ISF 264/23) and an equipment grant (ISF 532/23).

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

Table 4: Glossary and notation

| | |
|---|---|
| MLP | Multilayer Perceptron |
| CNN | Convolutional Neural Network |
| GNN | Graph Neural Network |
| GMN | Graph Metanetwork |
| INR | Implicit Neural Representation |
| $x, y, z, \cdots \in \mathbb{R}$ | Scalars |
| $\boldsymbol{x}, \boldsymbol{y}, \boldsymbol{z}, \cdots \in \mathbb{R}^n$ | Vectors |
| $\boldsymbol{X}, \boldsymbol{Y}, \boldsymbol{Z}, \cdots \in \mathbb{R}^{n \times m}$ | Matrices |
| $\mathcal{X}, \mathcal{Y}, \mathcal{Z}, \ldots$ | The domain of $\boldsymbol{x}, \boldsymbol{y}, \boldsymbol{z}, \ldots$ |
| $d_{\text{in}}, d_x, d_e, d_u \in \mathbb{N}$ | The size of the network input or node/edge/global features |
| $i, j \in \mathbb{N}$ | An index for edges |
| $(i, j)$ | A directed edge from index $j$ to $i$ |
| $\boldsymbol{P}$ | A permutation matrix |
| $\boldsymbol{W}$ | A weight matrix in a neural network |
| $\sigma$ | An element-wise nonlinearity function |
| $\boldsymbol{x} \in \mathcal{X} = \mathbb{R}^{d_{\text{in}}}$ | The input for a neural network |
| $\boldsymbol{\theta} \in \Theta$ | The parameters of an NN |
| $f_{\boldsymbol{\theta}}(\boldsymbol{x}) : \mathcal{X} \to \mathcal{X}'$ | A function parameterized by an neural network |
| $\tilde{\boldsymbol{\theta}} \in \Theta$ | A symmetry of $\boldsymbol{\theta}$, which induces the same $f$ |
| $V$ | The set of vertices |
| $E$ | The set of edges |
| $\mathcal{S}$ | Set of parameter-sharing constraints |
| $\text{DAG} = (V, E)$ | A Directed Acyclic Graph |
| $\phi : V \to V$ | An automorphism |
| $\boldsymbol{e} \in E$ | An edge |
| $\phi^{\boldsymbol{e}} : E \to E$ | The edge permutation for an automorphism |
| $\Phi : \Theta \to \Theta$ | The network parameter permutation induced by $\phi^{\boldsymbol{e}}$ |
| $\boldsymbol{e}_{(i,j)} \in \mathbb{R}^{d_e}$ | The features for the edge from $i$ to $j$ |
| $\boldsymbol{v}_i \in \mathbb{R}^{d_x}$ | The features for a node |
| $\boldsymbol{u} \in \mathbb{R}^{d_u}$ | A global feature |
| $\boldsymbol{h}_{\boldsymbol{v}}^{\boldsymbol{\theta}}(\boldsymbol{x}) \in \mathbb{R}$ | Activation at node $\boldsymbol{v}$ for $f_{\boldsymbol{\theta}}$ on input $\boldsymbol{x}$ |

## A  RELATED WORK

**Metanetworks.** Recently, several works developed metanets equivariant to permutation parameter symmetries of simple networks. DWSNets (Navon et al., 2023) and NFN (Zhou et al., 2023a) derive the form of linear layers that are group-equivariant to permutation parameter symmetries of simple MLPs (and NFN also handles CNNs). NFTs (Zhou et al., 2023b) use special parameter-permutation-equivariant attention layers and layer-number features to build Transformer-like metanets.

Other types of metanets have been developed, including those based on statistics such as weight mean and standard deviation (Unterthiner et al., 2020), 1D convolutional models on the flattened weight vector (Eilertsen et al., 2020), hierarchical LSTMs and per-parameter MLPs (Metz et al., 2022), and set functions applied to chunked parameter sets (Andreis et al., 2023).

Special metanets have also been developed to process implicit neural representations (INRs) representing shapes, images, or other types of continuous data (Dupont et al., 2022; Luigi et al., 2023; Bauer et al., 2023). However, these approaches require special training procedures for the input neural networks and/or access to forward passes of the input network. In contrast, graph metanets can process datasets of independently trained neural networks of diverse architectures without modifying the inputs or requiring forward passes through the inputs, as we demonstrate in our experiments.

The recent work of Zhang et al. (2023) also proposes graph neural networks as metanets. They use the computation graph construction of MLPs – not our general parameter graphs – which allows us to scale efficiently for models with parameter sharing (like CNNs). Moreover, they adopt a different graph construction for bias nodes, where our approach is inspired by expressive power considerations from the form of the parameter-permutation-equivariant linear maps (Navon et al.,

2023; Zhou et al., 2023a). Finally, Zhang et al. (2023) evaluate their method empirically on simple networks, but they have limited theoretical guarantees and empirical results on other networks.

**NNs as Graphs.** Many works consider neural networks as graphs, including early works that aimed at developing unified frameworks for neural networks (Bottou & Gallinari, 1990; Gegout et al., 1995). The graph perspective has been used in several application areas, including: neural architecture search (Liu et al., 2019; Xie et al., 2019), analyzing relationships between graph structure and performance (You et al., 2020), and federated learning with differing neural architectures (Litany et al., 2022). Several works process neural networks in some graph representation (Zhang et al., 2019; Thost & Chen, 2021; Knyazev et al., 2021; Litany et al., 2022), but these works generally operate on a much coarser network representation (e.g. where a node can correspond to a whole convolutional layer), whereas in our approach we desire each parameter to correspond to an edge.

**GNNs and Edge Representations.** There has been much work on graph neural networks (GNNs) in recent years, leading to many variants that can be used in our framework (Hamilton, 2020). GNNs learning edge representations instead of just node or whole-graph representations are particularly relevant. These include message-passing-like architectures, architectures based on graph-permutation-equivariant linear layers, and Transformers for graph data (Battaglia et al., 2018; Maron et al., 2019; Kim et al., 2021; 2022a;b; Diao & Loynd, 2023; Vignac et al., 2023; Ma et al., 2023).

## B    MORE GRAPH CONSTRUCTION DETAILS

We implemented a procedure to automatically build the parameter graph given a PyTorch (Paszke et al., 2017) definition of a model that uses supported layers. Our approach converts all layers in the network into their parameter subgraph representation and iteratively stitches the subgraphs together to form the overall parameter graph.

This section details how we build the per-layer subgraphs we partially described in Section 2.1. We also include additional examples in Figure 7, which we explain below.

**Node and edge features**    For all subgraph constructions, we use node and edge features to improve the expressive power of our GMNs. Some examples include layer index, neuron type (input neuron index, output neuron index, or layer type), edge direction (when message-passing in an undirected manner), and positional encodings (for parameters with associated spatial features, such as convolutions or the spatial feature grids detailed below). In all cases except for the residual layers, edge features always include the parameter value. All nodes and edges always include integer features for the layer index and neuron type. Importantly, these added features are invariant under neural DAG automorphisms, so adding them does not break the equivariance properties of our Graph Metanets.

**Linear layers**    For each linear layer, we use node and edge features corresponding to the layer index and the node/edge type. The latter is an integer value identifying that each node/edge is either a linear weight or bias type.

**Convolution layers**    For convolutions, we include additional edge features to identify each parameter's spatial position in the filter kernel. We use integer coordinates appended to the other edge features that indicate layer index and edge type. The node features include only the layer index and node type.

**Equivariant linear layers**    We can also handle general group-equivariant linear layers, of which convolutions are a special case. The high-level idea is the same: each layer has a node for each "channel", and each node in the input layer is connected with $d_b$ edges to each node in the output layer, where $d_b$ is the dimension of the space of equivariant linear maps.

More specifically, let the input space of the layer be $\mathbb{R}^{n_1}$, the output space be $\mathbb{R}^{n_2}$, and the symmetry group be $G$. Suppose the group acts via representations $\rho_1(g) \in \mathrm{GL}(n_1)$ on the input and $\rho_2(g) \in \mathrm{GL}(n_2)$ on the output. Then the set of all $G$-equivariant linear maps, i.e. the set of linear maps $T : \mathbb{R}^{n_1} \to \mathbb{R}^{n_2}$ such that $T \circ \rho_1(g) = \rho_2(g) \circ T$ for all $g \in G$, forms a vector space of dimension that we denote as $d_b$ (Maron et al., 2019; Finzi et al., 2021). Let $B_1, \ldots, B_{d_b}$ be a basis for the space

of equivariant linear maps. Then a group equivariant linear layer takes the form

$$x \mapsto \sum_{i=1}^{d_b} w_i B_i x,$$ (4)

for some learnable weights $w_i \in \mathbb{R}$. These weights $w_i$ are the parameters of the layer.

As is often done in practice, this can be extended to multiple input or output feature dimensions (also called channels) as follows. We now let the input space be $\mathbb{R}^{n_1 \times d_1}$ and the output space be $\mathbb{R}^{n_2 \times d_2}$, where $d_1$ is the number of input channels and $d_2$ the number of output channels. The group now acts independently on each channel: via representations $\tilde{\rho}_1(g) = \rho_1(g) \otimes I_{d_1 \times d_1}$ on the input, and $\tilde{\rho}_2(g) = \rho_2(g) \otimes I_{d_2 \times d_2}$ on the output; e.g. for an input matrix $X \in \mathbb{R}^{n_1 \times d_1}$, the action is given by $g \cdot X = \rho_1(g)X$. The basis of equivariant linear maps for this space is now has $d_b d_1 d_2$ elements, and they take the form $B_i \otimes E_{jl}$, where $i \in [d_b]$, $j \in [d_2]$, and $l \in [d_1]$. An equivariant linear layer then takes the form

$$X \mapsto \sum_{i=1}^{d_b} B_i X W^{(i)},$$ (5)

where $W_i \in \mathbb{R}^{d_1 \times d_2}$ are learnable weights. To construct a graph for this layer, we let the input layer have $d_1$ nodes, and the output layer have $d_2$ nodes. For $j \in [d_2]$ and $l \in [d_1]$, we connect input node $l$ to input node $j$ with $d_B$

Group-Equivariant Linear Layer (DeepSets)

Figure 6: Example of the graph construction for a group-equivariant linear map, for the specific case of the group $S_n$ of permutations acting on sets of $n$ elements in $\mathbb{R}^{n \times d}$ (Zaheer et al., 2017). There are two types of equivariant linear maps for this case. The depicted linear layer maps $\mathbb{R}^{n \times 2} \to \mathbb{R}^{n \times 3}$.

edges. The $i$th edge has as weight $W_{j,l}^{(i)}$. Moreover, the $i$th edge has a feature denoting that it is part of $B_i$ — in convolutions this is an encoding of the spatial position of the parameter in the kernel.

As an example, consider the permutation equivariant DeepSets linear layer (Zaheer et al., 2017). Here, the group $G$ of permutations on $n$ elements acts on $\mathbb{R}^n$ by permuting coordinates. There are two $G$-equivariant linear maps from $\mathbb{R}^n \to \mathbb{R}^n$, $I$ and $\mathbf{1}\mathbf{1}^\top$ (so $d_B = 2$). Thus, the equivariant linear layer from $\mathbb{R}^{n \times d_1} \to \mathbb{R}^{n \times d_2}$ takes the form

$$X \mapsto XW^{(1)} + \mathbf{1}\mathbf{1}^\top X W^{(2)}.$$ (6)

See Figure 6 for an illustration of how we encode this layer, as is used for our experiments in Section 4.1.

**Multi-head attention layers** For multi-head attention (Vaswani et al., 2017), we can include the head index as an additional node/edge feature for the relevant nodes and edges (though we do not currently do this in our experiments of Section 4.1, where we only use 2-head attention layers). Otherwise, we treat the edges as standard linear parameters.

**Residual layers** No parameters are associated with the edges for residual layers. Consequently, the edge features do not include the parameter values (we replace them with a constant value of 1). Otherwise, we include the layer index (for the starting layer) and an integer value identifying the edge as a residual connection.

**Normalization layers** Similar to bias nodes, normalization layers such as BatchNorm (Ioffe & Szegedy, 2015), LayerNorm (Ba et al., 2016), and GroupNorm (Wu & He, 2018) can be represented by two additional nodes corresponding to the learned mean and variance. Edges are drawn between these nodes and the neurons to be normalized. The middle of Figure 7 depicts a normalization layer with 3 neurons. We include the layer index and an integer value for both the node and edge features, identifying them as either the mean/variance and which normalization layer they correspond to. The different types of normalization layers are distinguished via node and edge features: for instance, LayerNorm mean parameter nodes are given a different node label from BatchNorm mean parameter nodes.

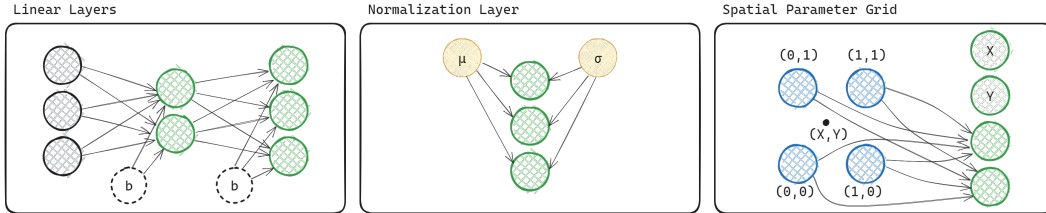

Figure 7: Additional examples of parameter subgraph constructions. We include linear layers in isolation for completeness (left), an example of a normalization layer (middle), and a spatial parameter grid (right).

**Spatial parameter grids**  An increasingly common design pattern in Implicit Neural Representations (INRs) introduces a grid of parameters that are accessed according to input coordinates, such as: triplanar grids (Chan et al., 2022), octrees (Takikawa et al., 2021), and hash grids (Müller et al., 2022). A typical implementation of these parameter grids involves finding a set of features according to the input coordinates and then using linear interpolation to combine the features. The resulting feature vector is then fed into an MLP to decode the features of a target object.

We show a subgraph construction for a simple 2D parameter grid in the right of Figure 7. The 2D grid contains only four entries at the corners of the domain, with two feature channels. We assume the two feature vectors are bilinearly interpolated based on the input coordinates $(X, Y)$. The resulting vector is concatenated with the input coordinates to produce a four-dimensional vector. In practice, we can add positional encoding of the grid position to the edge features for each element of the spatial parameter grid.

Note that the graph construction for the triplanar grid can largely be reused for dense voxel feature grids, or hash grids (Müller et al., 2022). The same principles also apply to sparse feature grids, for example, those using octrees (Takikawa et al., 2021). A recent work by Cardace et al. (2023) develops a Transformer-based metanet for processing INRs with triplanar grid features.

### B.1  CODE FOR CONSTRUCTING PARAMETER GRAPHS

We plan to release the code for constructing parameter graphs at a later date. Here, we outline some basics about how the code works. The code takes as input a PyTorch sequential module, which consists of several modules applied in sequence to some data. We iteratively build a graph, starting from the first layer in the input. For each layer in the input (e.g. a linear layer), we convert the layer to a subgraph, using the constructions outlined in Section 2.1 and Appendix B. We connect each of each of these subgraphs to the graph that we are iteratively building, until we have done this for all of the layers. We also have a way to invert this process, i.e. to go from parameter graph to neural network module. This is useful for metanet applications where we wish to modify the weights of an input neural network. To do this, we take an architecture specification and the edge weights of a parameter graph, and we then iteratively build a PyTorch module of the correct architecture and parameter values.

## C  THEORY: EQUIVARIANCE AND NEURAL GRAPH AUTOMORPHISMS

### C.1  DAGS AND COMPUTATION GRAPHS

Consider a DAG $(V, E)$, with weights $\boldsymbol{\theta} \in \mathbb{R}^{|E|}$ for each edge. We assume throughout that this DAG is connected. We may also have parameter-sharing constraints (e.g. in the case of convolution layers), which is a set $\mathcal{S}$ that partitions the edge set $E$, so that if edges $(i_1, j_1)$ and $(i_2, j_2)$ are in the same equivalence class, then $\boldsymbol{\theta}_{(i_1,j_1)} = \boldsymbol{\theta}_{(i_2,j_2)}$. Further choose a nonlinearity $\sigma : \mathbb{R} \to \mathbb{R}$. Let there be $d_{\text{in}}$ input nodes in a set $V_{\text{in}}$, $d_{\text{out}}$ output nodes in $V_{\text{out}}$, some number of bias nodes in $V_{\text{bias}}$, and every other node in a set $V_{\text{hidden}}$. This defines a neural network function $f_{\boldsymbol{\theta}} : \mathbb{R}^{d_{\text{in}}} \to \mathbb{R}^{d_{\text{out}}}$ with computation graph given by the DAG. To define the function, for an input $\boldsymbol{x} \in \mathbb{R}^{d_{\text{in}}}$ we first define

the *activation* of node $i \in V$ to be a real value $h_i^{\boldsymbol{\theta}}(\boldsymbol{x}) \in \mathbb{R}$, which is defined recursively as follows:

$$
h_i^{\boldsymbol{\theta}}(\boldsymbol{x}) = \begin{cases} \boldsymbol{x}_i & i \in V_{\text{in}} \\ 1 & i \in V_{\text{bias}} \\ \sigma \left( \sum_{(i,j) \in E} \boldsymbol{\theta}_{(i,j)} h_j^{\boldsymbol{\theta}}(\boldsymbol{x}) \right) & i \in V_{\text{hidden}} \\ \sum_{(i,j) \in E} \boldsymbol{\theta}_{(i,j)} h_j^{\boldsymbol{\theta}}(\boldsymbol{x}) & i \in V_{\text{out}} \end{cases} \tag{7}
$$

Note that the activations are well-defined because the graph is a DAG. Here, we slightly abuse notation and assume that nodes $1, \ldots, d_{\text{in}}$ are the input nodes. Finally, the neural network function is given by the values of the activations at the output nodes:

$$
f_{\boldsymbol{\theta}}(\boldsymbol{x}) = [h_{j_1}^{\boldsymbol{\theta}}(\boldsymbol{x}), \ldots, h_{j_{d_{\text{out}}}}^{\boldsymbol{\theta}}(\boldsymbol{x})] \in \mathbb{R}^{d_{\text{out}}}. \tag{8}
$$

Where $j_1, \ldots, j_{d_{\text{out}}} \in V_{\text{out}}$ are the output nodes. In the main paper, we generally consider our more efficient parameter graphs instead of computation graphs, since computation graphs may be much larger in general due to parameter sharing, in which one parameter is associated with many edges. Nonetheless, we study the theory of computation graphs here, as they are more closely related to the neural network function $f_{\boldsymbol{\theta}}$, and they can be defined unambiguously for feedforward neural networks (whereas we make some choices in our definitions of parameter graphs). We discuss extensions of our theoretical results to parameter networks in Appendix C.5.

## C.2 NEURAL DAG AUTOMORPHISMS

We define an automorphism (of Neural DAGs) to be a bijection $\phi : V \to V$ with the following three properties:

1. Edge preservation: $(\phi(i), \phi(j)) \in E$ if and only if $(i, j) \in E$.

2. Node label preservation: input nodes, output nodes, and bias nodes are fixed points, meaning they are mapped to themselves.

3. Weight sharing preservation: if $\boldsymbol{\theta}_{(i_1, j_1)}$ and $\boldsymbol{\theta}_{(i_2, j_2)}$ are constrained to be equal, then $\boldsymbol{\theta}_{(\phi(i_1), \phi(j_1))}$ and $\boldsymbol{\theta}_{(\phi(i_2), \phi(j_2))}$ are also constrained to be equal.

In particular, node label preservation means that if $i$ and $j$ are two distinct input, output, or bias nodes, then $\phi(i) = i$ and $\phi(j) = j$, and we cannot have $\phi(i) = j$.

## C.3 NEURAL DAG AUTOMORPHISMS GENERALIZE KNOWN PARAMETER SYMMETRIES

This result shows that every hidden neuron permutation in an MLP is an automorphism, and also that every automorphism of an MLP DAG takes this form.

**Proposition 5.** *A permutation $\phi : V \to V$ of an MLP DAG is a Neural DAG Automorphism if and only if $\phi$ only permutes hidden neurons, and $\phi(i)$ is in the same layer as $i$ for each hidden neuron $i$.*

*Proof.* ( $\Longrightarrow$ ) Suppose $\phi$ is a Neural DAG Automorphism. Then because of the node label preservation property, we know that $\phi$ maps each input neuron, output neuron, and bias neuron to itself, meaning that $\phi$ only permutes hidden neurons. Moreover, since $\phi$ is a graph isomorphism, it does not change path lengths: the maximum path length between $i$ and $j$ is the same as that between $\phi(i)$ and $\phi(j)$. Hence, $\phi$ preserves layer number of each hidden neuron.

( $\Longleftarrow$ ) Suppose $\phi$ only permutes hidden neurons and preserves the layer of each hidden neuron. We need only show edge preservation to show that $\phi$ is a Neural DAG Automorphism. If $(i, j) \in E$, then we know that $\text{layer}(j) = \text{layer}(i) - 1$, so $\text{layer}(\phi(j)) = \text{layer}(\phi(i)) - 1$. As MLPs are fully connected between adjacent layers, this means that $(\phi(i), \phi(j)) \in E$. Analogously, if $(\phi(i), \phi(j)) \in E$, then $\text{layer}(j) = \text{layer}(i) - 1$, so $(i, j) \in E$. We have thus showed edge preservation and are done. $\square$

For this next proposition, we will analyze the symmetries of simple CNN parameter spaces. In particular, we will assume the CNN only consists of convolutional layers (and no pooling, or fully connected layers). Every neuron has a spatial position and a channel index. For instance,

input neurons in a $32 \times 32$ input with 3 channels are indexed by a spatial position $(x, y) \in \{1, \ldots, 32\} \times \{1, \ldots, 32\}$, and a channel $c \in \{1, 2, 3\}$. We say that a permutation $\phi : V \to V$ *permutes hidden channels* if it only permutes hidden neurons, $\phi(i)$ has the same spatial position as $i$, and $\mathrm{channel}(i) = \mathrm{channel}(j)$ if and only if $\mathrm{channel}(\phi(i)) = \mathrm{channel}(\phi(j))$.

Here, we can show that any permutation of hidden channels that preserves neuron layers is a Neural DAG Automorphism, though we do not show the converse. Nonetheless, this direction shows that Neural DAG Automorphisms capture the known permutation parameter symmetries of CNNs, and thus our Graph Metanetwork framework can handle these symmetries.

**Proposition 6.** *Let $\phi : V \to V$ be a permutation of a CNN computation graph that permutes hidden channels and preserves neuron layers. Then $\phi$ is a Neural DAG Automorphism.*

*Proof.* Suppose $\phi$ only permutes hidden channels and preserves the layer of each hidden neuron. Then $\phi$ satisfies node label preservation, and also weight sharing preservation. Now, consider an edge $(i, j) \in E$, so $\mathrm{layer}(j) = \mathrm{layer}(i) - 1$, and also the spatial positions of $i$ and $j$ fit within the convolution filter of this layer. Since $\phi$ preserves layers, we have $\mathrm{layer}(j) = \mathrm{layer}(i) - 1$. Further, since $\phi$ only permutes hidden channels, the spatial position of $\phi(i)$ is the same as $i$, and likewise the spatial position of $\phi(j)$ is the same as $j$. Thus, $(\phi(i), \phi(j)) \in E$. A similar argument show that $(\phi(i), \phi(j))$ being in $E$ implies that $(i, j) \in E$, so we are done. $\square$

## C.4 NEURAL DAG AUTOMORPHISMS PRESERVE NETWORK FUNCTIONS

**Proposition 7.** *For any neural DAG automorphism $\phi$ of a computation graph, the neural network function is left unchanged: $\forall \boldsymbol{x} \in \mathcal{X}, f_{\boldsymbol{\theta}}(\boldsymbol{x}) = f_{\Phi(\boldsymbol{\theta})}(\boldsymbol{x})$.*

*Proof.* We will show a stronger statement: namely that the activations of the network with permuted parameters are permuted version of the activations of the original network. In other words, we will show that

$$\boldsymbol{h}_{\boldsymbol{v}}^{\boldsymbol{\theta}}(\boldsymbol{x}) = \boldsymbol{h}_{\phi(\boldsymbol{v})}^{\Phi(\boldsymbol{\theta})}(\boldsymbol{x}) \tag{9}$$

for all nodes $\boldsymbol{v} \in V$. This is sufficient to show the proposition for the following reason. Let the output nodes be $j_1, \ldots, j_{d_{\mathrm{out}}}$. Since a neural DAG automorphism $\phi$ maps each output node to itself, if equation 9 holds, then

$$f_{\Phi(\boldsymbol{\theta})}(\boldsymbol{x}) = [\boldsymbol{h}_{j_1}^{\Phi(\boldsymbol{\theta})}(\boldsymbol{x}), \ldots, \boldsymbol{h}_{j_{d_{\mathrm{out}}}}^{\Phi(\boldsymbol{\theta})}(\boldsymbol{x})] \tag{10}$$

$$= [\boldsymbol{h}_{\phi(j_1)}^{\Phi(\boldsymbol{\theta})}(\boldsymbol{x}), \ldots, \boldsymbol{h}_{\phi(j_{d_{\mathrm{out}}})}^{\Phi(\boldsymbol{\theta})}(\boldsymbol{x})] \tag{11}$$

$$= [\boldsymbol{h}_{j_1}^{\boldsymbol{\theta}}(\boldsymbol{x}), \ldots, \boldsymbol{h}_{j_{d_{\mathrm{out}}}}^{\boldsymbol{\theta}}(\boldsymbol{x})] \tag{12}$$

$$= f_{\boldsymbol{\theta}}(\boldsymbol{x}). \tag{13}$$

The second line follows as $\phi$ leaves the output nodes unchanged. The third line follows from equation 9. Thus, it suffices to show equation 9. We proceed by induction on the layer number of $\boldsymbol{v}$, i.e. the maximum path distance from an input node to $\boldsymbol{v}$.

For the base case, if $\boldsymbol{v}$ is an input node, then $\phi(\boldsymbol{v}) = \boldsymbol{v}$, so

$$\boldsymbol{h}_{\boldsymbol{v}}^{\boldsymbol{\theta}}(\boldsymbol{x}) = \boldsymbol{x}_{\boldsymbol{v}} = \boldsymbol{h}_{\boldsymbol{v}}^{\Phi(\boldsymbol{\theta})}(\boldsymbol{x}) = \boldsymbol{h}_{\phi(\boldsymbol{v})}^{\Phi(\boldsymbol{\theta})}(\boldsymbol{x}). \tag{14}$$

Bias nodes always have activation 1, so this combined with the fact that automorphisms map bias nodes to themselves show equation 9 for bias nodes.

Now, consider a hidden node $i$ at layer $l > 0$, and suppose that equation 9 holds for all nodes of lower layers. If $i$ is a hidden node, then we have

$$\boldsymbol{h}_{\phi(i)}^{\Phi(\boldsymbol{\theta})}(\boldsymbol{x}) = \sigma \left( \sum_{(\phi(i),\phi(j)) \in E} \Phi(\boldsymbol{\theta})_{(\phi(i),\phi(j))} \boldsymbol{h}_{\phi(j)}^{\Phi(\boldsymbol{\theta})}(\boldsymbol{x}) \right) \tag{15}$$

$$= \sigma \left( \sum_{(\phi(i),\phi(j)) \in E} \boldsymbol{\theta}_{(i,j)} \boldsymbol{h}_{\phi(j)}^{\Phi(\boldsymbol{\theta})}(\boldsymbol{x}) \right) \qquad \text{definition of } \Phi(\boldsymbol{\theta}) \tag{16}$$

$$= \sigma \left( \sum_{(\phi(i),\phi(j)) \in E} \boldsymbol{\theta}_{(i,j)} \boldsymbol{h}_{j}^{\boldsymbol{\theta}}(\boldsymbol{x}) \right) \qquad \text{induction} \tag{17}$$

$$= \sigma \left( \sum_{(i,j) \in E} \boldsymbol{\theta}_{(i,j)} \boldsymbol{h}_{j}^{\boldsymbol{\theta}}(\boldsymbol{x}) \right) \qquad \text{edge preservation property of } \phi \tag{18}$$

$$= \boldsymbol{h}_{i}^{\boldsymbol{\theta}}(\boldsymbol{x}). \tag{19}$$

If $i$ is an output node, then the same derivation holds without the nonlinearity $\sigma$. Thus, we have shown equation 9, and we are done. $\qquad \square$

## C.5 FROM COMPUTATION GRAPHS TO PARAMETER GRAPHS

We conjecture that Proposition 1 can be extended to automorphisms over parameter graphs instead of computation graphs. Here we discuss the formal intuition behind this conjecture.

To prove that the network function remains unchanged after an automorphism of the parameter graph, we could show that any automorphism of the parameter graph corresponds to an automorphism of the computation graph. One way to do this is through a surjective homomorphism from the computation graph to the parameter graph that is locally-injective (Fiala & Kratochvíl, 2008). This requires that each 1-neighbourhood in the computation graph is mapped injectively to a 1-neighbourhood in the parameter graph.

Intuitively, this allows us to define an automorphism on the parameter graph and lift the automorphism to the computation graph through the preimage of the homomorphism. Concretely, we apply the parameter graph automorphism to the computation graph by permuting the preimage sets for each node in the parameter graph under the homomorphism.

Using this machinery, we expect that the proof of Proposition 1 can be extended to parameter graphs. One point of difficulty is that our parameter graph construction allows for multigraphs — which are valuable for succinct representations and also make it easier to stitch together subgraphs.

## C.6 EQUIVARIANCE OF GRAPH METANETWORKS TO NEURAL DAG AUTOMORPHISMS

Here, we show that graph metanetworks are equivariant to architecture graph symmetries. In particular, consider an architecture graph $G = (V, E)$. We also allow modifications to the graph, so that $G$ does not need to be the exact DAG of the computation graph; for instance, $G$ can be the undirected version of the computation graph. Consider a fixed ordering of the $n$ nodes, and let $X \in \mathbb{R}^{n \times d_x}$ be the input node features, and $A \in \mathbb{R}^{n \times n \times d_e}$ be the input edge features. For any permutation $\eta : V \to V$, we define the standard group action: $\eta(X) \in \mathbb{R}^{n \times d_x}$ is defined by $\eta(X)_{\eta(i),k} = X_{i,k}$, and $\eta(A) \in \mathbb{R}^{n \times n \times d_e}$ is defined by $\eta(A)_{\eta(i),\eta(j),k} = A_{i,j,k}$.

Consider a graph function that updates node features and edge features, say $\text{GNN} : \mathbb{R}^{n \times d_x} \times \mathbb{R}^{n \times n \times d_e} \to \mathbb{R}^{n \times d_x} \times \mathbb{R}^{n \times n \times d_e}$. For inputs $X$ and $A$, let $\text{GNN}(X, A)^X \in \mathbb{R}^{n \times d_x}$ denote the updated node features, and $\text{GNN}(X, A)^A \in \mathbb{R}^{n \times n \times d_e}$ denote the updated edge features. We say such a function is permutation equivariant if for all permutations $\eta : [n] \to [n]$, it holds that

$$\text{GNN}(\eta(X), \eta(A))_{\eta(i),k}^{X} = \text{GNN}(X, A)_{i,k}^{X} \tag{20}$$

$$\text{GNN}(\eta(X), \eta(A))_{\eta(i),\eta(j),k}^{A} = \text{GNN}(X, A)_{i,j,k}^{A}. \tag{21}$$

Graph neural networks have exactly this permutation equivariance (Maron et al., 2019). Thus, since any architecture graph symmetry $\tau$ is a permutation $[n] \to [n]$, it holds that graph metanetworks are equivariant to architecture graph symmetries. Thus, we restate our result:

**Proposition 8.** *Graph metanets are equivariant to parameter permutations induced by neural DAG automorphisms.*

However, this would also seem to imply that graph metanetworks are equivariant to more permutations than just architecture graph symmetries, which would be undesirable because this would mean reduced expressivity. The key to fixing this problem is via adding the additional node and edge features described in Appendix B. Recall that we choose additional node and edge features, say $\tilde{X}$ and $\tilde{A}$, that are invariant to neural DAG automorphisms. That is, $\phi(\tilde{X}) = \tilde{X}$ and $\phi(\tilde{A}) = \tilde{A}$ for any neural DAG automorphism $\phi$. Also, we add these features in a fixed manner, so they do not depend on the choice of node indices. Thus, the output node features of the Graph Metanet on the original graph are $\mathrm{GNN}([X, \tilde{X}], [A, \tilde{A}])^X$, whereas after applying a permutation they are $\mathrm{GNN}([\eta(X), \tilde{X}], [\eta(A), \tilde{A}])^X$. These outputs are in general not equal up to a permutation. However, when $\eta$ is a neural DAG automorphism, they are equal, because

$$\mathrm{GNN}([\phi(X), \tilde{X}], [\phi(A), \tilde{A}])^X_{\eta(i),k} \tag{22}$$

$$= \mathrm{GNN}([\phi(X), \phi(\tilde{X})], [\phi(A), \phi(\tilde{A})])^X_{\eta(i),k} \qquad \tilde{X}, \tilde{A} \text{ invariant to } \phi \tag{23}$$

$$= \mathrm{GNN}([X, \tilde{X}], [A, \tilde{A}])^X_{i,k} \qquad \text{GNN equivariance.} \tag{24}$$

The same argument holds for output edge features.

## D  THEORY: EXPRESSIVE POWER OF GRAPH METANETWORKS

**Representation vs. Approximation.**  In this Appendix section, we prove results on the expressive power of Graph Metanets. For these results, there is some continuous target function $f_{\mathrm{target}}$ that we wish to approximate (either StatNN, NP-NFN, or forward passes through a neural network). In our proofs, for ease of exposition we show that if we assume that the MLPs of our Graph Metanets can be arbitrary continuous functions, then there is some Graph Metanet that is exactly $f_{\mathrm{target}}$.

However, MLPs cannot generally express every continuous function. Instead, MLPs are known to be universal approximators of continuous functions (Hornik, 1991), meaning that for any continuous $f_{\mathrm{target}}$ with inputs from a compact domain, for every precision $\epsilon > 0$ there is an MLP that is within $\epsilon$ (in some function-space metric) to $f_{\mathrm{target}}$. We will call any function class with this property a universal function class. If our metanets were just a single MLP, then it would be trivial to go from an exact representation statement to an approximation statement. However, in general our metanets are compositions of several MLPs and other operations.

Despite this discrepancy, we can appeal to standard techniques that have been used in recent works in the study of expressive power of equivariant neural networks to show approximation results (Lim et al., 2023; Navon et al., 2023). In particular, if a neural network architecture can be written as a composition of functions $f_L \circ \cdots \circ f_1$, and each of the functions comes from a universal function class, then the composition also forms a universal function class. Thus, one need only show that each component can be approximated to show that an entire target function can be approximated. This is captured in the following lemma, which is proved in Lim et al. (2023).

**Lemma 1.** *Let $\mathcal{X} \subseteq \mathbb{R}^{d_0}$ be compact, and let $\mathcal{F}_1, \ldots, \mathcal{F}_L$ be families of continuous functions, where $\mathcal{F}_i$ consists of functions from $\mathbb{R}^{d_{i-1}} \to \mathbb{R}^{d_i}$. Let $\mathcal{F}$ be the set of compositions of these functions, so $\mathcal{F} = \{f_L \circ \cdots \circ f_1 : \mathcal{X} \to \mathbb{R}^{d_L}, f_i \in \mathcal{F}_i\}$.*

*For each $i$, let $\tilde{\mathcal{F}}_i$ be a family of continuous functions that universally approximates $\mathcal{F}_i$. Then the family of compositions $\tilde{\mathcal{F}} = \{\tilde{f}_L \circ \cdots \circ \tilde{f}_1 : \tilde{f}_i \in \tilde{\mathcal{F}}_i\}$ universally approximates $\mathcal{F}$.*

### D.1  PROOF: SIMULATING EXISTING METANETWORKS

#### D.1.1  SIMULATING STATNN

In its most expressive form, StatNN (Unterthiner et al., 2020) computes statistics from each weight and bias of an MLP or CNN input, and then applies an MLP on top of these statistics. The

statistics considered by Unterthiner et al. (2020) are the mean, variance, and $q$th percentiles for $q \in \{0, 25, 50, 75, 100\}$. These are all continuous permutation invariant functions of the input, i.e. for any of these statistics $s$ and flattened weight vector $w$ (e.g. $w = \text{vec}(W^{(l)})$ for a weight matrix $W^{(l)}$ or $w = b^{(l)}$ for a bias vector $b^{(l)}$), it holds that $s(w) = s(Pw)$ for any permutation matrix $P$. This is the key to showing that Graph Metanets can express StatNN, since the global graph feature update can compute these permutation invariant functions, and then the final invariant layer can apply an MLP on the global graph feature.

**Proposition 9.** *Graph Metanetworks can express StatNN (Unterthiner et al., 2020) on MLP input networks.*

*Proof.* Let $W^{(1)}, \ldots, W^{(L)}$ be the weights and $b^{(1)}, \ldots, b^{(L)}$ the biases of an input MLP. Let $s$ be a continuous permutation invariant function with output in $\mathbb{R}^{d_s}$, and let $\text{MLP}^{\text{StatNN}} : \mathbb{R}^{2Ld_s} \to \mathbb{R}$ be the MLP of the StatNN, so

$$\text{StatNN}(W^{(1)}, \ldots, W^{(L)}, b^{(1)}, \ldots, b^{(L)}) = \text{MLP}^{\text{StatNN}}(s(W^{(1)}), \ldots, s(b^{(L)})). \quad (25)$$

We will show that a Graph Metanetwork can express this function. Let the edge features $\boldsymbol{e}_{(i,j)} \in \mathbb{R}^3$ be three dimensional, with the first dimension holding the parameter value, the second dimension denoting the layer index, and the third dimension denoting whether the edge is associated to a weight or bias parameter. We will not require node features for this proof. Thus, our Graph Metanetwork will take the form

$$\boldsymbol{e}_{(i,j)} \leftarrow \text{MLP}^{\boldsymbol{e}}(\boldsymbol{e}_{(i,j)}) \quad (26)$$

$$\boldsymbol{u} \leftarrow \text{MLP}^{\boldsymbol{u}}\left(\sum_{\boldsymbol{e} \in E} \boldsymbol{e}\right), \quad (27)$$

and the result will be that $\boldsymbol{u} \in \mathbb{R}$ will be the desired StatNN output.

Since $s$ is a continuous permutation invariant function, it can be expressed in the form $s(w) = \rho(\sum_i \phi(w_i))$ for continuous functions $\rho : \mathbb{R}^{d_\phi} \to \mathbb{R}^{d_s}$ and $\phi : \mathbb{R} \to \mathbb{R}^{d_\phi}$ (Zaheer et al., 2017). Let $\text{MLP}^{\boldsymbol{e}} : \mathbb{R}^3 \to \mathbb{R}^{2L \times d_\phi}$ be given by

$$[\text{MLP}^{\boldsymbol{e}}(\boldsymbol{e}_{(i,j)})]^{l,:} = \phi(\boldsymbol{e}_{(i,j)}) \cdot \mathbb{1}\left[[\boldsymbol{e}_{(i,j)}]^2 = l, [\boldsymbol{e}_{(i,j)}]^3 = \texttt{weight}\right] \quad (28)$$

$$[\text{MLP}^{\boldsymbol{e}}(\boldsymbol{e}_{(i,j)})]^{l+L,:} = \phi(\boldsymbol{e}_{(i,j)}) \cdot \mathbb{1}\left[[\boldsymbol{e}_{(i,j)}]^2 = l, [\boldsymbol{e}_{(i,j)}]^3 = \texttt{bias}\right] \quad (29)$$

for $l = 1, \ldots, L$. Here, $[a]^i$ is the $i$th entry of the vector $a$, $[A]^{i,:}$ denotes the $i$th entry of the matrix $A$, and $\mathbb{1}$ is one if the condition is true and zero otherwise. Note that we write the output of $\text{MLP}^{\boldsymbol{e}}$ as a matrix to make indexing simpler, but it functionally is just a vector.

Then note that the input to $\text{MLP}^{\boldsymbol{u}}$ is $\sum_{\boldsymbol{e} \in E} \boldsymbol{e}$, which is a matrix $A \in \mathbb{R}^{2L \times d_\phi}$ that satisfies:

$$[A]^{l,:} = \sum_i \phi(\text{vec}(W^{(l)})_i) \quad (30)$$

$$[A]^{l+L,:} = \sum_i \phi(\text{vec}(b^{(l)})_i). \quad (31)$$

Hence, we have that

$$\rho([A]^{2l,:}) = s(W^{(l)}) \quad (32)$$

$$\rho([A]^{2l+1,:}) = s(b^{(l)}). \quad (33)$$

Thus, we can let $\text{MLP}^{\boldsymbol{u}} : \mathbb{R}^{2L \times d_\phi} \to \mathbb{R}$ be given by

$$\text{MLP}^{\boldsymbol{u}}(A) = \text{MLP}^{\text{StatNN}}\left(\rho([A]^{1,:}), \ldots, \rho([A]^{2L,:})\right) \quad (34)$$

and we are done. $\qquad\square$

### D.1.2 SIMULATING NP-NFN

**Proposition 10.** *Graph Metanetworks can express NP-NFN (Zhou et al., 2023a) on MLP input networks.*

*Proof.* **GNN Form.** Here is the form of our GNN, where for simplicity of the proof, we move the global graph feature update before the other updates and remove one of the node-update MLPs. Note that the $\text{MLP}^{\boldsymbol{u}}$ is inside the sum — this is important for our construction.

$$\boldsymbol{u} \leftarrow \sum_{(i,j)\in E} \text{MLP}^{\boldsymbol{u}}\left(\boldsymbol{e}_{(i,j)}, \boldsymbol{u}\right) \tag{35}$$

$$\boldsymbol{v}_i \leftarrow \sum_{j:(i,j)\in E} \text{MLP}^{\boldsymbol{v}}(\boldsymbol{v}_i, \boldsymbol{v}_j, \boldsymbol{e}_{(i,j)}, \boldsymbol{u}) \tag{36}$$

$$\boldsymbol{e}_{(i,j)} \leftarrow \text{MLP}^{\boldsymbol{e}}(\boldsymbol{v}_i, \boldsymbol{v}_j, \boldsymbol{e}_{(i,j)}, \boldsymbol{u}). \tag{37}$$

**Graph Form.** We make the graph undirected by including the reverse edge for each edge in the DAG and labelling these as backward edges. We assume the nodes and edges are endowed with certain features at the start (the global graph feature $u$ is initialized to 0). Each node has the features: layer number and node type (input neuron number, output neuron number, hidden neuron, or bias neuron number). Each edge has the features: weight value, layer number, weight type, whether it is backward or not, and whether it is a weight or bias edge. Notably, each of these features is invariant to neural graph automorphisms. The node and edge features belong to the following sets:

$$\boldsymbol{v}_i \in \begin{bmatrix} \{0, \ldots, L\} \\ \{\texttt{in}_1, \ldots, \texttt{in}_{d_{\text{in}}}, \texttt{out}_1, \ldots, \texttt{out}_{d_{\text{out}}}, \texttt{bias}_1, \ldots, \texttt{bias}_L, \texttt{hidden}\} \end{bmatrix} \tag{38}$$

$$\boldsymbol{e}_{(i,j)} \in \begin{bmatrix} \mathbb{R} \\ \{1, \ldots, L\} \\ \{\texttt{forward}, \texttt{backward}\} \\ \{\texttt{weight}, \texttt{bias}\} \end{bmatrix} \tag{39}$$

**Without biases.** For illustration, we will first prove this result for the case when we are only processing weights and no biases. First, we recall the form of the NP-NFN linear layer (Zhou et al., 2023a). It takes MLP weights $W \in \mathcal{W} = \mathbb{R}^{d_L \times d_{L-1}} \times \ldots \times \mathbb{R}^{d_1 \times d_0}$, and outputs new weight representations $H(W) \in \mathcal{W}$. We let $W^{(l)}$ denote the weight matrix of the $l$th layer, and let $W_{i,j}^{(l)}$ be the $(i,j)$th entry of this matrix. Then, the NP-NFN linear layer takes the form:

$$H(W)_{i,j}^{(l)} = \left(\sum_{s=1}^{L} a_1^{l,s} W_{\star,\star}^{(s)}\right) + a_2^{l,l} W_{\star,j}^{(l)} + a_3^{l,l-1} W_{j,\star}^{(l-1)} \tag{40}$$

$$+ a_4^{l,l} W_{i,\star}^{(l)} + a_5^{l,l+1} W_{\star,i}^{(l+1)} + a_6^{l} W_{i,j}^{(l)}, \tag{41}$$

where we define $W^{(0)} = 0$ and $W^{(L+1)} = 0$ for the boundary cases of $l = 1$ and $l = L$. Here, the $a \in \mathbb{R}$ are learnable scalars, and the $\star$ denotes a summation over all relevant indices. For instance, for $W^{(1)} \in \mathbb{R}^{n_1 \times n_0}$, we have $W_{\star,k}^{(1)} = \sum_{i=1}^{n_1} W_{i,k}^{(1)}$ and $W_{\star,\star}^{(1)} = \sum_{i=1}^{n_1} \sum_{j=1}^{n_0} W_{i,j}^{(1)}$.

First, we use the global graph feature to compute the first summands. Let $\text{MLP}^{\boldsymbol{u}}$ have an output space of $\mathbb{R}^L$, and let

$$[\text{MLP}^{\boldsymbol{u}}(\boldsymbol{e}, \boldsymbol{u})]^l = [\boldsymbol{e}]^1 \cdot \mathbb{1}[[\boldsymbol{e}]^2 = l], \tag{42}$$

where $[y]^i \in \mathbb{R}$ is the $i$th entry of the vector $y$, and $\mathbb{1}$ is one if the condition inside is true, and zero otherwise. Since the first column of $\boldsymbol{e}$ holds the weight and the second column holds the layer number, the global graph feature is updated as

$$\boldsymbol{u} \leftarrow \begin{bmatrix} W_{\star,\star}^{(1)} & \ldots & W_{\star,\star}^{(L)} \end{bmatrix}^{\top} \in \mathbb{R}^L. \tag{43}$$

Next, we consider the node feature updates. We define $\text{MLP}^{\boldsymbol{v}}$ to have an output space of $\mathbb{R}^2$, such that

$$[\text{MLP}^{\boldsymbol{v}}(\boldsymbol{v}_i, \boldsymbol{v}_j, \boldsymbol{e}_{(i,j)}, \boldsymbol{u})]^1 = [\boldsymbol{e}_{(i,j)}]^1 \cdot \mathbb{1}[[\boldsymbol{e}_{(i,j)}]^3 = \texttt{backward}] \tag{44}$$

$$[\text{MLP}^{\boldsymbol{v}}(\boldsymbol{v}_i, \boldsymbol{v}_j, \boldsymbol{e}_{(i,j)}, \boldsymbol{u})]^2 = [\boldsymbol{e}_{(i,j)}]^1 \cdot \mathbb{1}[[\boldsymbol{e}_{(i,j)}]^3 = \texttt{forward}]. \tag{45}$$

For a node $v$ that is the $k$th node in layer $l$ (where the input nodes are at layer 0), the node update is then given by:

$$v \leftarrow \begin{bmatrix} W^{(l+1)}_{\star,k} & W^{(l)}_{k,\star} \end{bmatrix} \in \mathbb{R}^2. \tag{46}$$

Finally, we define the edge update. We let $\mathrm{MLP}^e$ have an output space of $\mathbb{R}$, such that for any forward edge $(i,j)$ the update is (writing $l = [e_{(i,j)}]^2$ for brevity):

$$\mathrm{MLP}^e(v_i, v_j, e_{(i,j)}, u) = \tag{47}$$

$$\sum_{s=1}^{L} a_1^{l,s}[u]^s + a_2^{l,l}[v_j]^1 + a_3^{l,l-1}[v_j]^2 + a_4^{l,l}[v_i]^2 + a_5^{l,l+1}[v_i]^1 + a_6^l[e_{(i,j)}]^1. \tag{48}$$

By plugging in the above forms of $u$, $v_i$, and $v_j$ (and noting that $[e_{(i,j)}]^1 = W^{(l)}_{i,j}$), we see that this new edge representation is exactly the NP-NFN update, so we are done.

**With biases.** Recall the form of the NP-NFN linear layer where the input MLP also has biases:

$$H(W, b) = (\tilde{W}, \tilde{b}) \tag{49}$$

$$\tilde{W}^{(l)}_{i,j} = \left( \sum_{s=1}^{L} a_1^{l,s} W^{(s)}_{\star,\star} \right) + a_2^{l,l} W^{(l)}_{\star,j} + a_3^{l,l-1} W^{(l-1)}_{j,\star} \tag{50}$$

$$+ a_4^{l,l} W^{(l)}_{i,\star} + a_5^{l,l+1} W^{(l+1)}_{\star,i} + a_6^l W^{(l)}_{i,j} \tag{51}$$

$$+ \left( \sum_{s=1}^{L} a_7^{l,s} b^{(s)}_\star \right) + a_8^l b^{(l)}_i + a_9^l b^{(l-1)}_j \tag{52}$$

$$\tilde{b}^{(l)}_j = \left( \sum_{s=1}^{L} c_1^{l,s} W^{(s)}_{\star,\star} \right) + c_2^{l,l} W^{(l)}_{j,\star} + c_3^{l,l+1} W^{(l+1)}_{\star,j} \tag{53}$$

$$+ \left( \sum_{s=1}^{L} c_4^{l,s} b^{(s)}_\star \right) + c_5^l b^{(l)}_j \tag{54}$$

Note that we define $b^{(0)} = 0$, and once again define $W^{(0)} = 0$ and $W^{(L+1)} = 0$ for the boundary cases.

We define the global model $\mathrm{MLP}^u$ to have output in $\mathbb{R}^{2L}$, such that for $l \in \{1, \ldots, L\}$,

$$[\mathrm{MLP}^u(v, e, u)]^l = [e]^1 \cdot \mathbb{1}\left[[e]^2 = l, [e]^4 = \texttt{weight}\right] \tag{55}$$

$$[\mathrm{MLP}^u(e, u)]^{l+L} = [e]^1 \cdot \mathbb{1}\left[[e]^2 = l, [e]^4 = \texttt{bias}\right], \tag{56}$$

The global graph feature then takes the form

$$u \leftarrow \begin{bmatrix} W^{(1)}_{\star,\star} & \ldots & W^{(L)}_{\star,\star} & b^{(1)}_\star & \ldots & b^{(L)}_\star \end{bmatrix}^\top \in \mathbb{R}^{2L}. \tag{57}$$

Next, we consider the node feature updates. We define $\mathrm{MLP}^v$ to have an output space of $\mathbb{R}^3$, such that

$$[\mathrm{MLP}^v(v_i, v_j, e_{(i,j)}, u)]^1 = [e_{(i,j)}]^1 \cdot \mathbb{1}\left[[e_{(i,j)}]^3 = \texttt{backward}, [e_{(i,j)}]^4 = \texttt{weight}\right] \tag{58}$$

$$[\mathrm{MLP}^v(v_i, v_j, e_{(i,j)}, u)]^2 = [e_{(i,j)}]^1 \cdot \mathbb{1}\left[[e_{(i,j)}]^3 = \texttt{forward}, [e_{(i,j)}]^4 = \texttt{weight}\right] \tag{59}$$

$$[\mathrm{MLP}^v(v_i, v_j, e_{(i,j)}, u)]^3 = [e_{(i,j)}]^1 \cdot \mathbb{1}\left[[e_{(i,j)}]^3 = \texttt{forward}, [e_{(i,j)}]^4 = \texttt{bias}\right] \tag{60}$$

For a non-bias node $v$ that is the $k$th node in layer $l$, the node update is then given by:

$$v \leftarrow \begin{bmatrix} W^{(l+1)}_{\star,k} & W^{(l)}_{k,\star} & b^{(l)}_k \end{bmatrix} \in \mathbb{R}^3. \tag{61}$$

Finally, we define the edge update. We let $\mathrm{MLP}^e$ have an output space of $\mathbb{R}$, such that for any forward edge $(i, j)$ the update is (writing $l = [\boldsymbol{e}_{(i,j)}]^2$ for brevity):

$$\mathrm{MLP}^e(\boldsymbol{v}_i, \boldsymbol{v}_j, \boldsymbol{e}_{(i,j)}, \boldsymbol{u}) = \tag{62}$$

$$\mathbb{1}\left[[\boldsymbol{e}_{(i,j)}^4] = \mathtt{weight}\right]\left(\sum_{s=1}^{L} a_1^{l,s}[\boldsymbol{u}]^s + a_2^{l,l}[\boldsymbol{v}_j]^1 + a_3^{l,l-1}[\boldsymbol{v}_j]^2\right. \tag{63}$$

$$+ a_4^{l,l}[\boldsymbol{v}_i]^2 + a_5^{l,l+1}[\boldsymbol{v}_i]^1 + a_6^l[\boldsymbol{e}_{(i,j)}]^1 \tag{64}$$

$$\left.+ \sum_{s=1}^{L} a_7^{l,s}[\boldsymbol{u}]^{s+L} + a_8^l[\boldsymbol{v}_i]^3 + a_9^l[\boldsymbol{v}_j]^3\right) \tag{65}$$

$$+\mathbb{1}\left[[\boldsymbol{e}_{(i,j)}]^4 = \mathtt{bias}\right]\left(\sum_{s=1}^{L} c_1^{l,s}[\boldsymbol{u}]^s + c_2^{l,l}[\boldsymbol{v}_j]^2 + c_3^{l,l+1}[\boldsymbol{v}_j]^1\right. \tag{66}$$

$$\left.+ \sum_{s=1}^{L} c_4^{l,s}[\boldsymbol{u}]^{s+L} + c_5^l[\boldsymbol{e}_{(i,j)}]^1\right) \tag{67}$$

By plugging in the above forms of $\boldsymbol{u}$, $\boldsymbol{v}_i$, and $\boldsymbol{v}_j$, we see that this new edge representation is exactly the NP-NFN update, so we are done. $\qquad\square$

### D.1.3 Expressive Power with Bias Nodes

Here, we explain how encoding biases in linear layers as bias nodes can aid expressive power (in the simple case of MLPs). Intuitively, this is because the bias node allows for different bias parameters of the same layer to communicate with each other in each layer of a message passing graph metanet, whereas this is not as simple and requires at least two layers if biases are encoded as self-loops or node features.

For a specific example, when the graph metanet does not use a global feature (as in our experiments), then the bias node can allow message passing between biases within the same layer. In the NP-NFN case covered in Appendix D.1.2, we see that the global feature allows message passing between biases of any layer. When there is no global feature, letting $\boldsymbol{v}_{b^{(l)}}$ denote the bias node of the $l$th layer, then our graph metanet can nonetheless update

$$\boldsymbol{v}_{b^{(l)}} \leftarrow \sum_{i=1}^{d_l} b_i^{(l)} = b_\star^{(l)} \tag{68}$$

$$\boldsymbol{e}_{(b^{(l)},i)} \leftarrow c\boldsymbol{v}_{b^{(l)}} = cb_\star^{(l)} \tag{69}$$

for some scalar $c \in \mathbb{R}$. This is one of the equivariant linear maps included in NP-NFN. On the other hand, a graph metanet with no global feature cannot compute this in one layer if the biases are encoded as self-loops or node features — this would require at least two layers.

### D.2 Proof: Simulating Forward Passes

**Proposition 11.** *On computation graph inputs, graph metanets can express the forward pass of any input feedforward neural network as defined in Section 2.2.*

*Proof.* We will prove this statement by showing that a graph metanet can compute the activations $\boldsymbol{h}_v^{\boldsymbol{\theta}}(\boldsymbol{x})$ for an input network $f_{\boldsymbol{\theta}}$ and input $\boldsymbol{x}$. In particular, we will show that $l$ layers of a graph metanet can compute the activation $\boldsymbol{h}_v^{\boldsymbol{\theta}}(\boldsymbol{x})$ for any node $v$ with layer number at most $l$, where the layer number is defined as the maximum path distance from any input node to $v$. We do this by induction on $l$. For the base case, $\boldsymbol{h}_v^{\boldsymbol{\theta}}(\boldsymbol{x}) = \boldsymbol{x}_v$ for any input node $v$, so at depth zero of the graph metanet the induction hypothesis is satisfied. Also, the activation for a bias node is always 1, which a graph metanet can compute using a node-update MLP that outputs 1 for any node with the bias-node label.

For a hidden node $i$ with layer number $l > 0$, recall that we have (output nodes are handled similarly):

$$h_i^{\boldsymbol{\theta}}(\boldsymbol{x}) = \sigma \left( \sum_{(i,j) \in E} \boldsymbol{\theta}_{(i,j)} h_j^{\boldsymbol{\theta}}(\boldsymbol{x}) \right). \tag{70}$$

Let $\boldsymbol{v}_i^{(l-1)}$ denote the node feature computed for node $i$ at depth $l-1$ of the graph metanet. By induction, we have $\boldsymbol{v}_j^{(l-1)} = h_j^{\boldsymbol{\theta}}(\boldsymbol{x})$ for any node $j$ in layer $l-1$ or earlier. Recall the form of the node update in a graph metanet layer:

$$\boldsymbol{v}_i \leftarrow \mathrm{MLP}_2^{\boldsymbol{v}} \left( \boldsymbol{v}_i, \sum_{(i,j) \in E} \mathrm{MLP}_1^{\boldsymbol{v}}(\boldsymbol{v}_i, \boldsymbol{v}_j, \boldsymbol{e}_{(i,j)}, \boldsymbol{u}), \boldsymbol{u} \right) \tag{71}$$

Let $\mathrm{MLP}_1^{\boldsymbol{v}}$ for depth $l$ of the GNN be given by

$$\mathrm{MLP}_1^{\boldsymbol{v}}(\boldsymbol{v}_i, \boldsymbol{v}_j, \boldsymbol{e}_{(i,j)}, u) = [\boldsymbol{e}_{(i,j)}]^1 \boldsymbol{v}_j, \tag{72}$$

and let $\mathrm{MLP}_2^{\boldsymbol{v}}$ for depth $l$ be given by

$$\mathrm{MLP}_2^{\boldsymbol{v}}(\boldsymbol{v}_i, a, \boldsymbol{u}) = \sigma(a). \tag{73}$$

Then the node update equation for the GNN is

$$\boldsymbol{v}_i^{(l)} = \sigma \left( \sum_{(i,j) \in E} \boldsymbol{\theta}_{(i,j)} \boldsymbol{v}_j^{(l-1)} \right) \tag{74}$$

$$= \sigma \left( \sum_{(i,j) \in E} \boldsymbol{\theta}_{(i,j)} h_j^{\boldsymbol{\theta}}(\boldsymbol{x}) \right) \tag{75}$$

$$= h_i^{\boldsymbol{\theta}}(\boldsymbol{x}). \tag{76}$$

The same derivation holds for an output node $i$, replacing $\sigma$ with the identity map. So, we are done by induction. $\square$

# E  ADDITIONAL EXPERIMENTAL DETAILS

## E.1  PREDICTING ACCURACY

### E.1.1  CLASSIFIER TRAINING DETAILS

For the predicting accuracy experiments, we train two datasets of about $30\,000$ CIFAR-10 image classification neural networks each. We use the Fast Forward Computer Vision (FFCV) library (Leclerc et al., 2023) for fast training and build off their template training setup for CIFAR-10. We use random horizontal flips, random translations, and Cutout (DeVries & Taylor, 2017) as data augmentation. The learning rate schedule has a linear warmup and then a linear decay to zero. All the other hyperparameters are given in Table 5; we sample many of the hyperparameters for each neural network. For the Diverse Architectures dataset, we train about 6000 of each of the 5 types of networks, while for the Varying CNNs dataset we train about $30\,000$ of the 2D-CNNs.

### E.1.2  METANETWORK TRAINING DETAILS

**Data splits.** For the 50% training data experiment, we take a random split of $15\,000$ training networks, and for the 5% training data experiment we take a random split of 1500 training networks. Then 2000 random networks are selected for validation, and the rest are for testing.

For the OOD experiments, we only train and validate on networks of the lowest hidden dimension of each architecture. This means that we only train and validate on CNNs (both 2D and 1D) of hidden dimension 24, DeepSets of hidden dimension 32, ResNets of hidden dimension 16, and ViTs of hidden dimension 32. 2000 of these shallower networks are chosen for validation, and the rest for training. Then we test on all networks with higher hidden dimension.

Table 5: Hyperparameters for the CIFAR-10 image classifiers that we trained for the predicting accuracy experiments in Section 4.1.

| Model | Hyperparameter | Distribution / Choice |
|---|---|---|
| All | Learning rate | $10^{-\mathrm{Uniform}(1,3)}$ |
| | Weight decay | $10^{-\mathrm{Uniform}(2,5)}$ |
| | Label smoothing | $\mathrm{Uniform}(0, .2)$ |
| | Optimizer | $\mathrm{Uniform}(\{\mathrm{SGD}, \mathrm{Adam}, \mathrm{RMSprop}\})$ |
| CNN (2D) | Hidden dimension | $\mathrm{Uniform}(\{24, 28, 32\})$ |
| | # Convolution layers | $\mathrm{Uniform}(\{1, 2, 3\})$ |
| | # Linear layers | $\mathrm{Uniform}(\{1, 2\})$ |
| | Normalization type | $\mathrm{Uniform}(\{\mathrm{BatchNorm}, \mathrm{GroupNorm}\})$ |
| | Dropout | $\mathrm{Uniform}(0, .25)$ |
| | Kernel size | 3 |
| CNN (1D) | Hidden dimension | $\mathrm{Uniform}(\{24, 28, 32\})$ |
| | # Convolution layers | $\mathrm{Uniform}(\{1, 2, 3\})$ |
| | # Linear layers | $\mathrm{Uniform}(\{1, 2\})$ |
| | Normalization type | $\mathrm{Uniform}(\{\mathrm{BatchNorm}, \mathrm{GroupNorm}\})$ |
| | Dropout | $\mathrm{Uniform}(0, .25)$ |
| | Kernel size | 9 |
| DeepSets | Hidden dimension | $\mathrm{Uniform}(\{32, 64\})$ |
| | # Equivariant layers | $\mathrm{Uniform}(\{2, 3, 4\})$ |
| | # Linear layers | $\mathrm{Uniform}(\{1, 2\})$ |
| | Normalization type | $\mathrm{Uniform}(\{\mathrm{BatchNorm}, \mathrm{GroupNorm}\})$ |
| | Dropout | $\mathrm{Uniform}(0, .25)$ |
| ResNet | Hidden dimension | $\mathrm{Uniform}(\{16, 32\})$ |
| | # Blocks | $\mathrm{Uniform}(\{2, 3, 4\})$ |
| | # Linear layers | 1 |
| | Normalization type | BatchNorm |
| | Dropout | 0 |
| ViT | Hidden dimension | $\mathrm{Uniform}(\{32, 48\})$ |
| | # Transformer blocks | $\mathrm{Uniform}(\{2, 3\})$ |
| | # Linear layers | 1 |
| | Normalization type | LayerNorm |
| | Dropout | $\mathrm{Uniform}(0, .25)$ |

Table 6: Additional results for predicting accuracy on the dataset of small CNNs from Unterthiner et al. (2020). We see that GMNs outperform all metanetworks. Here, we use the Graph Transformer GRIT (Ma et al., 2023) as our graph learning model for GMN, as we find it performs better than the message passing models that we tried.

| Metanetwork | Test $\tau$ |
|---|---|
| NFN-HNP | $.934\pm.001$ |
| NFN-NP | $.922\pm.001$ |
| StatNN | $.915\pm.002$ |
| NFT | $.926\pm.001$ |
| GMN (ours) | $\mathbf{.938}\pm.000$ |

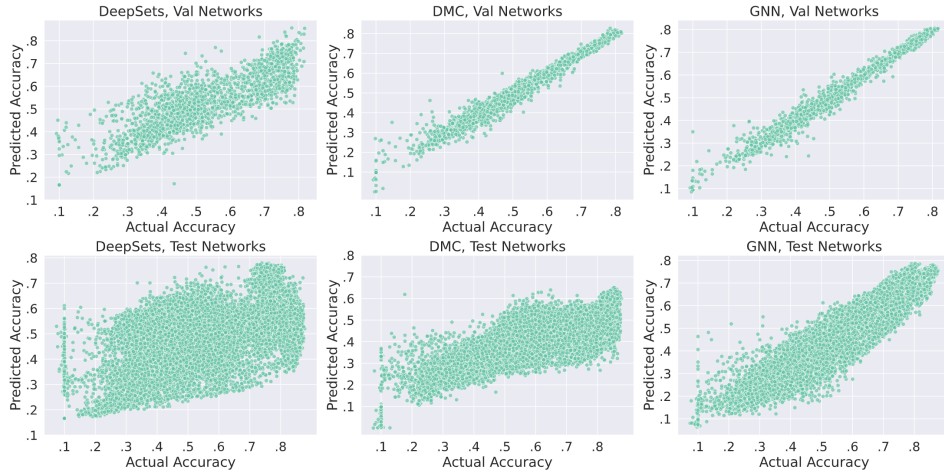

Figure 8: Scatterplots of actual test accuracies (x-axis) and metanetwork-predicted test accuracies (y-axis) for image classifiers from the Diverse Architectures dataset with the OOD train/test split. Top row shows validation split input networks, and bottom row shows test split input networks.

**Metanetworks.** For each metanet, we first choose hyperparameters such that the total number of trainable parameters is around $750\,000$. Then we search learning rates $\alpha \in \{.0001, .0005, .001, .005, .01\}$, and choose the learning rate that achieves the best validation $R^2$. Finally, we train the model using this best learning rate with 5 random seeds and report the mean and standard deviation of the test set $R^2$ and Kendall $\tau$ values.

We train all metanets with the Adam optimizer (Kingma & Ba, 2014). The training loss is a binary cross entropy loss between the predicted and true accuracy; each metanet has a sigmoid nonlinearity at the end to ensure that its input is within $[0, 1]$.

**Small CNN Experiments** To compare against previously proposed metanetworks, we also run experiments on predicting accuracy of the small CNNs trained by Unterthiner et al. (2020), following the experimental setup of Zhou et al. (2023a). Results are given in Table E.1.2. We see that our GMNs outperform all other metanetworks. We use GRIT (Ma et al., 2023) as our graph learning model, which is a type of Graph Transformer. Our GRIT model has $3\,652\,865$ parameters, uses a hidden dimension of 256, has 4 layers, is trained with a learning rate of `3e-4`, and uses 16 heads in its attention modules.

### E.2 EDITING 2D INRS DETAILS

We closely follow the experimental setup of Zhou et al. (2023a) for the experiments in Section 4.2. As in their work, we train the metanets for $50\,000$ iterations with a batch size of 32, using the Adam optimizer with .001 learning rate. We follow their setup to parameterize the updated weights $\tilde{\theta}$ as

$\tilde{\boldsymbol{\theta}} = \boldsymbol{\theta} + \gamma \cdot \mathrm{Metanet}(\boldsymbol{\theta})$, where $\boldsymbol{\theta}$ are the parameters of the INR representing the unedited image, and $\gamma$ are learned scalars for each parameter that are initialized to .01. Our message passing GNN metanetworks have $8\,945\,671$ parameters, hidden dimension of 512, and 4 message passing layers.

### E.3 SELF-SUPERVISED LEARNING WITH INRS DETAILS

We closely follow the experimental setup of Navon et al. (2023) for the self-supervised learning task. In particular, as they do in their experiments, we choose hyperparameters for our graph metanet such that the number of trainable metanetwork parameters is about $100\,000$. Thus, we use a graph metanetwork with 3 GNN layers and a hidden dimension of 76.

We follow the same training procedure as Navon et al. (2023), which uses a contrastive objective with data augmentations given by adding Gaussian noise to the weights and masking weights to zero. As in Navon et al. (2023), we evaluate our metanet on three random seeds, and report the mean and standard deviation across these three runs.

