# OpenReview forum: "Graph Metanetworks for Processing Diverse Neural Architectures"
_ICLR.cc/2024/Conference — ICLR 2024 spotlight_

### Official Review · Reviewer_fJUi · 2023-10-27

**Soundness:** 3 good
**Presentation:** 4 excellent
**Contribution:** 3 good
**Rating:** 6
**Confidence:** 3

**Summary:**

The paper proposed a new approach to represent neural networks and their weights using a so called parameter graph in which nodes represent neurons and edges represent weights/parameters of the neural network. The paper demonstrates how to build this graph for different kinds of architectures including non-trivial and very practical cases such as transformers. The paper formally shows useful properties of the approach such as permutation equivariance corresponding to neuron equivariance. Finally, the results in two experiments show better performance than the baselines.

**Strengths:**

The paper has several good contributions:

1. The paper addresses an interesting and promising topic of representing neural networks and their weights.
2. The proposed parameter graph is a reasonable approach and looks much simpler than previous works such as DWSNets and NFN/NFT. It is similar to the concurrent work of Zhang et al. (2023), which is properly credited.
3. Description of how to build graphs for different layer types such as convolution, self-attention and residual connections is very informative.
4. The experiments show reasonably good results of the proposed approach.

The paper is also well written and organized.

**Weaknesses:**

The paper have several weaknesses. I'm willing to revise the rating based on authors' response.

1. The paper says that "While our graphs are DAGs, we are free to use undirected edges". Would not the direction of edges be a useful feature in some cases? For example, sometimes networks take multiple inputs and have multiple outputs so there is no way to differentiate input vs output unless edge direction is used.

2. The computational complexity vs other approaches is not analyzed. Can the model encode large models in a feasible way?

3. There are very few experiments. The authors could have more experiments following previous works from DWSNets, NFN, etc. or be creative in designing more novel experiments. For example, generation of new networks mentioned in the intro (Erkoc¸ et al., 2023) could be a very appealing experiment.

4. The experiments lack ablations to understand how the model behaves under different settings. These could be the number of layers/params in the graph metanet, different GNN architectures, different approaches to treat the bias term (e.g. comparing to Zhang et al. (2023)), ablating different components of the GNN in 2.3, etc.

5. Source code is not available in the submission, which would be very helpful at least for how to build a graph given a neural network. Do the authors intend to open source the code?

6. The purpose of Section 3 and Proposition 2 is a bit unclear, these could be replaced by more experiments or computational complexity analysis that are lacking.

7. Section 5.2. lacks details. How the vector representation is obtained? Details of training the GNN are missing. More difficult INR tasks could be added.

**Questions:**

1. Given that the same graph metanet can process diverse architectures, can the model trained in 5.1 be directly applied to large realistic models like ResNet-50 or large ViTs? For example, in the paper "Zero-Cost Proxies for Lightweight NAS" there are PyTorchCV networks with recorded accuracies. It would be very interesting to see how the results would look like for such challenging cases. Moreover, in Section 5.1 it would be interesting to see correlation performance of the methods from "Zero-Cost Proxies for Lightweight NAS" like gradnorm, which are very easy to compute. Another interesting experiment could be to track the predicted accuracy during training some network and see if it correlates well with the actual accuracy. Given that evaluation of large networks is very expensive, this approach could be an alternative way to track performance.

2. Is it correct that the trained metanets cannot generalize to some architectures, for example to the kernels of larger size than seen during training because the edges would have more features? If yes, this should be clearly described in Limitations or another appropriate section.

---

> ### Author Response · Authors · 2023-11-18
> **Response 1/2**
>
> > “1. ... Would not the direction of edges be a useful feature in some cases? For example, sometimes networks take multiple inputs and have multiple outputs so there is no way to differentiate input vs output unless edge direction is used. “
>
> We add layer number (of both edges and nodes) as features for the input; this is mentioned in Appendix A. This allows the network to differentiate input and output nodes. Also, we can add edge direction as a feature (forward or backward), which is mentioned in Appendix A and found to be useful in the expressive power results in Appendix C.1.2.
>
> > “2. The computational complexity vs other approaches is not analyzed. Can the model encode large models in a feasible way?”
>
> As message passing GNNs are generally efficient, and as much work has been done on GNNs, our methodology inherits these benefits. While we have not yet considered tasks on large networks, we note that GNNs are capable of training on graphs with 300 million edges (and hence networks with 300 million parameters) on a single GPU with 24 GB of RAM in 32-bit precision (e.g. https://arxiv.org/pdf/2110.14446.pdf). Thus, using multiple GPUs, GPUs with more RAM, and standard memory-saving tricks such as lower precision and gradient checkpointing, we believe that GMNs process input networks with billions of parameters. For inference, GMNs can ostensibly process input networks with tens of billions of parameters, due to lower memory cost than training.
>
> > “3. There are very few experiments. The authors could have more experiments following previous works from DWSNets, NFN, etc. or be creative in designing more novel experiments…”
>
> As explained in the general comment, we have added an additional set of experiments in our revised manuscript, including a 2D-INR editing task. This includes a comparison with NFN and the more recent NFT method [Zhou et al. 2023b].
>
> We agree that there are many exciting applications for this approach, especially given our ability to handle heterogeneous architectures within a single trained model. While we hope to explore these, we emphasize that our current submission contains significant contributions in terms of the novel method that we proposed, our theoretical analysis of it, and the empirical experiments validating the method in practice.
>
> > “4. The experiments lack ablations to understand how the model behaves under different settings…”
>
> We agree that an ablation study would improve the empirical rigor of our work. We are working on this and hope to include it soon. Speaking informally, we found our metanets to be robust and effective across different choices of the GNN architecture, and choices of feature encoding.
>
> > “5. Source code is not available in the submission, which would be very helpful at least for how to build a graph given a neural network. Do the authors intend to open source the code?”
>
> Unfortunately, we are unable to provide source code at this time. We are in the process of preparing an open-source release, and hope to have this ready in time for the camera-ready publication. *In the meantime, we have added high level descriptions for our graph constructions in Appendix A.1.*
>
> > “6. The purpose of Section 3 and Proposition 2 is a bit unclear, these could be replaced by more experiments or computational complexity analysis that are lacking.”
>
> We are disappointed that this contribution of our work was not clearer. We emphasize that these results are extensions/generalizations of established theorems presented in prior published work. Additionally, these results theoretically justify our application of graph neural networks as metanets.
>
> While we certainly agree that more experiments could strengthen our paper, and we have added more since submission, we stand by the inclusion of these theoretical results as a major contribution.
>
> See our general comment for more on the value of our theory.

---

> > ### Author Response · Authors · 2023-11-18
> > **Response 2/2**
> >
> > > “7. Section 5.2. lacks details. How the vector representation is obtained? Details of training the GNN are missing. More difficult INR tasks could be added.”
> >
> > We agree with the reviewer and apologize for the oversight. We have added more detail to this section and the associated Appendix. We have also now included an INR editing task (see general response).
> >
> > > “1. Given that the same graph metanet can process diverse architectures, can the model trained in 5.1 be directly applied to large realistic models like ResNet-50 or large ViTs? For example, in the paper "Zero-Cost Proxies for Lightweight NAS" there are PyTorchCV networks with recorded accuracies. It would be very interesting to see how the results would look like for such challenging cases. Moreover, in Section 5.1 it would be interesting to see correlation performance of the methods from "Zero-Cost Proxies for Lightweight NAS" like gradnorm, which are very easy to compute. Another interesting experiment could be to track the predicted accuracy during training some network and see if it correlates well with the actual accuracy. Given that evaluation of large networks is very expensive, this approach could be an alternative way to track performance.”
> >
> > Given that in Section 5.1 we show that the same graph metanet can predict the accuracy of small ResNets and ViTs, we think our models can likely be useful for larger models. We agree with the reviewer that these larger scale accuracy prediction tasks could be interesting and useful. In some sense, future versions of our metanetwork may allow predictions of new types of “scaling laws” for input neural nets.
> >
> > > “2. Is it correct that the trained metanets cannot generalize to some architectures, for example to the kernels of larger size than seen during training because the edges would have more features? ...”
> >
> > While we don’t establish a concrete answer to the specific question of kernel size, we do not have evidence that our method cannot generalize to novel architectures. We rely on GNNs, which can be applied to any input graph. Consequently, we can train and evaluate our metanets on vastly different parameter graphs, including graphs with kernel sizes never seen during training.
> >
> > To handle the specific case of kernel size, we use positional encoding of the kernel location as edge features. One could have the center kernel position encoded as (0,0) and then encode arbitrary kernel sizes by moving radially outwards from the center when computing positional encoding. This would allow our model to handle different kernels at training and testing time. Note that we do not establish empirical generalization to this particular distribution shift. However, we do show that our metanets generalize to a similar distribution shift: increasing network width.

---

> > > ### Comment · Reviewer_fJUi · 2023-11-22
> > >
> > > I appreciate authors' response which clarified some issues but also revealed other potential issues.
> > > Additional 2D INR results are useful but indicating standard deviation would be beneficial, otherwise it's hard to compare numbers. Intentions to open-source the code are very welcome especially given additional concerns expressed below.
> > >
> > > **Node and edge features**
> > >
> > > Appendix A shows useful details of the graph construction, but the paper's main text would benefit from taking into account these details. For example, having in the main text "we are free to use undirected edges" while actually using "edge direction" in some results sounds misleading.
> > > It's unclear if the node/edge features (described in Appendix A) compromise the equivariance properties of the metanet (proved in the main text) or generalization capabilities (e.g. OOD in Table 1)?
> > > - For example "input neuron index, output neuron index" should be explained more. It sounds like the indices are fixed for all input networks so the metanet outputs would be different depending on the neuron permutation.
> > > It's also unclear how the metanet can generalize to wider networks given that neuron indices will be unseen (larger).
> > > Also the "Node and edge features" paragraph is vague saying "Some examples include", so it remains mysterious if there are other node/edge features used.
> > > - Using "layer number" might be problematic too. For example, for networks with residual connections the residual branches are parallel (i.e. do not have any order), so it's unclear what kind of "layer number" is used in this paper and whether these features will preserve equivariance properties or not. Also the layer number features will probably prevent generalization to deeper networks?
> > > - There are "layer type" features, however what kind of types are predefined is unclear. Are there are only weight and bias type or other types like conv layer / norm layer / etc. Does using layer types imply that the metanet cannot generalize to unseen layer types?
> > >
> > > Without ablations showing how node/features affect the results and how they affect equivariance, it's hard to understand what exactly makes the model more effective than the baselines (e.g. can these features improve the baselines? how much?).
> > > The response does not address this concern effectively. For example, saying that some features are "useful in the expressive power results" while at the same time saying that "our metanets to be robust and effective across different choices of the GNN architecture, and choices of feature encoding.".
> > >
> > > Given these concerns, the limitations and generalization capabilities of the proposed metanet could be communicated more clearly.
> > >
> > > Regarding "Section 3" the contributions are understood and are reasonable. The purpose was unclear in a sense that more experiments/ablations and details of the method in the main text would be more appreciated.

---

> ### Author Response · Authors · 2023-11-22
>
> Thank you for your response. We greatly appreciate you taking the time to read our rebuttal and paper in further detail.
>
> > Additional 2D INR results are useful but indicating standard deviation would be beneficial
>
> Apologies for this oversight. We will of course include standard deviations in the final version, as we have with other empirical results.
>
>
> Thank you for your detailed questions about the feature constructions that we use. This has highlighted that this part of the paper deserves further clarification, and we will certainly do so in a later revision. In general, none of these features are strictly necessary and simply serve the purpose of improving network expressivity without sacrificing equivariance. Here are brief responses to the specific points that you raised:
>
> > having in the main text "we are free to use undirected edges" while actually using "edge direction" in some results sounds misleading
>
> We will clarify this in the main text. Certainly, our proposed approach works with and without edge direction feature encoding. Some of our theoretical results lean on edge direction features (without violating equivariance), but empirically we found that this is unnecessary.
>
> > It sounds like the indices are fixed for all input networks so the metanet outputs would be different depending on the neuron permutation
>
> (We will assume MLP input networks for ease of exposition, but the same argument can be generalized across all architectures.)
>
> To clarify, neuron indices are not included for hidden neurons. We expose only the neuron indices for the input layer and output layer as these should not be included in the desired permutation equivariance. We assume that the input neurons come in some pre-fixed permutation, which is the case in typical learning settings. We (and prior work) are then only concerned with hidden neuron permutations. However, input/output permutations could also be trivially included if desired (by not exposing their neuron indices).
>
> Wider networks (meaning more hidden units) are possible as the network cannot differentiate between any two neurons in the same hidden layer. This becomes a standard(-ish) application of GNNs generalizing to novel in-distribution graphs. We verify this empirically in our paper.
>
> > Using "layer number" might be problematic too
>
> Including the layer number is not necessary, but is helpful for empirical performance. Residual edges can be encoded using the layer index of the outgoing nodes (each residual connection certainly has an output neuron generating it). Layer number is identical for all neurons within a single layer, and thus these cannot be distinguished by the GNN, and equivariance is retained. For what it's worth, this same general idea was independently proposed in the concurrent work of Zhang et al., where learned positional encodings are used as features to identify each layer. We found that the simpler layer index was sufficient.
>
> We agree that generalization to deeper networks is an interesting question. There are several ways that this could be tackled. For example, one could try encoding the layer index as a continuous value from 0->1 with 0 for the input layer and 1 for the output layer. Or indeed, you could remove layer index entirely as a feature and rely on the GNN to learn this via message passing.
>
> > There are "layer type" features, however what kind of types are predefined is unclear.
>
> We include layer type features for all unique layer types that we can represent (e.g. linear, convolution, multi-head attention, etc.). The features are simply categorical variables, with each layer type (in some predefined order) being given values in increments of 1. More sophisticated choices could be explored.
>
> If a novel layer type was provided at inference time, then we would not generally expect our metanets to generalize well when these features are being used. But this is intuitive, as the network was never trained on networks with these layer types. However, one could forego the layer-type features entirely --- paying the cost of reduced expressiveness for improved generalization.
>
> We agree that further empirical investigation would help to build an increased understanding of our metanets. But we would like to emphasize again that we have presented a novel and effective approach to designing equivariant metanetworks. While we will continue to strengthen our empirical evaluation by adding ablations and other improvements, the existing results show consistently high performance of our method on several tasks, alongside valuable new capabilities that other methods cannot achieve: generalization to different input architectures and handling diverse and heterogeneous input networks.

---

### Official Review · Reviewer_ffRA · 2023-11-01

**Soundness:** 3 good
**Presentation:** 3 good
**Contribution:** 2 fair
**Rating:** 6
**Confidence:** 4

**Summary:**

This paper introduces a novel approach called Graph Metanetworks (GMNs) that utilizes graph neural networks to process diverse neural architectures. The authors address the challenge of generalising metanetwork architectures to different types of networks by building graphs representing the input neural networks and processing them using graph neural networks. The GMNs are proven to be expressive and equivariant to parameter permutation symmetries, and they demonstrate superior performance on various metanetwork tasks across different neural network architectures.

**Strengths:**

The proposed GMNs can generalise to different types of neural architectures. Unlike previous works e.g. Navon et al. (2023) that were tailored to specific networks, the GMNs can handle a wide range of architectures, including those with complex modules such as attention blocks. Upon the main idea of processing weights with graph networks in Zhang et al. (2023), this paper extends the method to a variety of neural layers that are common in modern neural architectures.

The authors also provide a proof of the expressiveness and equivariance of GMNs to parameter permutation symmetries, which is an important property for metanetworks.

**Weaknesses:**

- Although different neural layers and architectures can be encoded as the proposed parameter graph representation, however, the information in the spatial domain is missing (e.g. translation equivariance and receptive field in ConvNets). This could be an inherent and general limitation of the proposed method as well as other related work on weight domain.

- There are also a variety of non-parametric operations in feed-forward neural networks that are not addressed by the paper. E.g. pooling layers (especially max pooling), atrous convolutions, padding and so on. I can expect these operations can be encoded as some additional indicating features but I wonder if there are better and less artificial solutions.

- Comparison with Navon et al. (2023) only on the 1D sine curve toy dataset, but not on the more challenging 2D INR image tasks in the original paper. Other experiments take DeepSets and DMC as baselines which are 2017 and 2020 papers.

**Questions:**

- Since graph can encode a variety of architectures, I wonder if it is possible to generalise and especially extrapolate prediction to unseen architecture, e.g. the test MLP/CNN is wider or deeper than all training examples.

- I do not see pooling layers discusses. Are there pooling layers involved in the networks in the dataset?

- How to encode atrous/dilated convolutions?

---

> ### Author Response · Authors · 2023-11-18
>
> > “Although different neural layers and architectures can be encoded as the proposed parameter graph representation, however, the information in the spatial domain is missing (e.g. translation equivariance and receptive field in ConvNets). This could be an inherent and general limitation of the proposed method as well as other related work on weight domain.”
>
> We are not sure what the reviewer means here, a clarification may be helpful. For instance, the receptive field is readily available in the parameter graph representation, since the kernel sizes are encoded, so any metanetwork can trivially determine the receptive field. Notably, we include node and edge type features, so for instance a node corresponding to a convolutional layer channel has a node feature indicating that it is a convolution node.
>
>
> > “There are also a variety of non-parametric operations in feed-forward neural networks that are not addressed by the paper. E.g. pooling layers (especially max pooling), atrous convolutions, padding and so on. I can expect these operations can be encoded as some additional indicating features but I wonder if there are better and less artificial solutions.”
>
> Our method is able to represent a wide variety of network architectures while retaining expressivity. **We already handle many more operations than previous work**, e.g. NFN and DWSNets only handle linear or convolutional layers (and not normalization, residuals, attention, group equivariant linear maps, or spatial parameter grids as we handle).  This is achieved through a combination of parameter graph design and feature encoding.
>
> For instance, with regards to spatial pooling layers, these are essentially a no-op in the parameter graph. In the sense that the same parameter graph accurately captures the equivariances that we’re seeking whether pooling is included or not. In fact, the image classifiers that we process in Section 5.1 have a global mean pooling (we have added this note to the revision).
>
> Following our current design philosophy, adding features is exactly the right solution to indicate to the GNN that a pooling layer has been applied. It is also effectively free in terms of computation cost. We are not sure what the reviewer means when they say that adding features is “artificial”, but we note that adding features is very efficient (essentially zero cost), general, and simple.
>
>
> > “Comparison with Navon et al. (2023) only on the 1D sine curve toy dataset, but not on the more challenging 2D INR image tasks in the original paper. Other experiments take DeepSets and DMC as baselines which are 2017 and 2020 papers.”
>
> As noted in the general comment, we now include experiments on 2D INR editing tasks from [Zhou et al. 2023a]. Our method outperforms the NFN baseline (NFN-HNP is essentially the same approach as DWSNets) but is beaten by the more recent specialized NFT architecture on only one of the tasks. We expect this gap can be closed by using more powerful/sophisticated graph neural network architectures, but emphasize at this time that our more general approach is applicable across a wide range of input networks while NFN/NFT are designed only for simple MLPs and simple CNNs (without e.g. normalization layers or residual connections).
>
> We have also included additional results using NFN and NFT to predict generalization on a fixed CNN architecture. Our more general approach outperforms or matches the performance of these specialized baselines.
>
> > “Since graph can encode a variety of architectures, I wonder if it is possible to generalize and especially extrapolate prediction to unseen architecture, e.g. the test MLP/CNN is wider or deeper than all training examples.”
>
> **Our initial submission already had experiments** that evaluate the capability of our approach to generalize to unseen architectures. Please see the “OOD” setting in Section 5.1, and Figure 5. Notably, our GNNs generalize well to wider networks than those seen in training, for all of the diverse architectures: 2D CNN, 1D CNN, DeepSets, ResNet, ViT.
>
> > “I do not see pooling layers discusses. Are there pooling layers involved in the networks in the dataset?”
>
> Yes, we use mean pooling across the spatial dimensions before the fully connected prediction layers for the image classifiers in Section 5.1. We have noted this in the revision.
>
> > “How to encode atrous/dilated convolutions?”
>
> One way to do this is to view the atrous/dilated convolution as a larger sparse kernel in the standard way, with zeros in many entries. E.g. a 3x3 kernel with dilation 2 can be viewed as a 5x5 kernel with zeros in all but 9 entries.
>
> Another, more efficient way to encode these is to add another edge feature indicating the dilation factor for the convolutional weights. E.g. standard convolutions get an additional edge feature of 1, while convolution with dilation factor $k$ gives an edge feature of $k$.

---

> > ### Comment · Reviewer_ffRA · 2023-11-20
> >
> > Thank you for the response.
> >
> > To clarify my first point and give an example, convolution layers incorporate some spatial inductive bias with its filter and parameter sharing paradigm. However with the proposed graph representation, first the convolution filter is flattened to 1D features so the 2D spatial relationship of filter weights is lost, which might be geometrically meaningful (e.g. can identify gradients at a specific 2D direction).

---

> > > ### Author Response · Authors · 2023-11-21
> > >
> > > Thank you for quickly clarifying! We see what you mean now.
> > >
> > > However, we do not think this is necessarily much of an issue. To recap, for each weight in a convolution filter, we include an edge between the nodes representing the input and output channel. This edge is given a feature that is a positional encoding of the (x,y) spatial location of the weight in the original filter. Then we use a sinusoidal positional encoding for this (x,y) feature. Note that sinusoidal positional encodings like this are used in Vision Transformers, as they also take 2D grids, flatten them, and then add in 2D information again via (x,y) positional encodings. As Vision Transformers and similar approaches are of course very successful, this approach of encoding spatial information via positional encodings is probably very reasonable.
> > >
> > > If we want to include 2D spatial information more explicitly, then we can take another approach using 2D convolutions in our metanetwork. For instance, if we have a 5x5 convolution filter, this would give 25 edges between each input and output channel. The edge features of these edges can be themselves obtained by a standard 2D convolution network applied on the 5x5 filter, which more explicitly accounts for 2D spatial information. After this, our standard graph metanet framework can be followed, with further processing done by a GNN.

---

### Official Review · Reviewer_ZWAE · 2023-11-02

**Soundness:** 3 good
**Presentation:** 3 good
**Contribution:** 3 good
**Rating:** 6
**Confidence:** 3

**Summary:**

The idea is to construct directed acyclic graphs (DAGs) that are also ‘parameter graphs’ (which represent parameters as weighted edges) that represent neural networks and feed them through a simple message-passing graph neural network ‘metanetwork’. According to them, they are distinguished from other works that do this by a few design choices of the graph construction. They show that the way they construct neural network DAGs is invariant to the order in which neurons on the same layer (ones between which order should not matter) are embedded, which is stated to be an issue with the way some other metanets represent networks. They state how they represent different kinds of layers in their ‘parameter graphs’, and show that it can represent layers others cannot (such as normalization layers, according to them) and have less scaling issues with parameter-sharing layers such as convolution and attention layers. They evaluate by attempting to learn the prediction accuracy of datasets of image classifier networks on CIFAR-10, one is of 2d CNNS, and one is of varying models (CNNs, ViTs, ResNets, etc) with competing methods (those which can represent the inputs).

**Strengths:**

It should be noted that I am not familiar with other papers regarding metanets and am basing my assessment largely on information from this paper. That being said, given that they represent the current state of this area fairly, this seems to be an impressive paper. They appear to address scaling issues with similarly expressive network representations and expand the variety of representable networks, which seem to be great contributions.

**Weaknesses:**

They address most of my concerns I had while reading, including some tests to compare against the newer state of the art metanets mentioned that were excluded from most of the result due to not being able to represent certain types of layers. Notably a cited competing method ‘NFN’ was left out from this, though seems to be addressed in the appendix, and as it apparently deals exclusively with MLPs I believe it is fair not to compare with the proposed method.

**Questions:**

Please see weakness.

---

> ### Author Response · Authors · 2023-11-18
>
> > “They address most of my concerns I had while reading, including some tests to compare against the newer state of the art metanets mentioned that were excluded from most of the result due to not being able to represent certain types of layers. Notably a cited competing method ‘NFN’ was left out from this, though seems to be addressed in the appendix, and as it apparently deals exclusively with MLPs I believe it is fair not to compare with the proposed method.”
>
> We appreciate the author’s positive review. We have now run additional experiments on simple input networks (simple CNNs and MLPs) to compare our method against both NFN and the more recent NFT [Zhou et al. 2023b]; see the general comment for the results. In summary, our Graph Metanetworks generally outperform or match existing metanetworks on these tasks with simple input networks. Further, as you mention, our Graph Metanetworks can process significantly more diverse and complex input networks than these methods.
>
> We hope that this addresses any remaining concerns, and would appreciate the opportunity to answer any other questions that you have.

---

### Author Response · Authors · 2023-11-18
**General Comment to Reviewers**

We thank the reviewers for their time and effort spent reviewing our paper. We have uploaded a revised PDF, which we feel improves the submission and addresses the reviewers’ concerns. Here, we would like to address two important points common to the reviews.
1. First, we discuss our theoretical contribution.
2. Second, we include two new sets of experiments, which include another application not present in our original submission (2D neural implicit editing), and also comparisons to recent methods NFN [Zhou et al. 2023a] and NFT [Zhou et al. 2023b].

**Theory**

Since the reviewers did not comment much on, or questioned the value of our theoretical results, we will motivate them more clearly here. In prior work. DWSNets and NFN showed that equivariance to parameter symmetries gives SOTA results, but their models only work for very simple architectures. Our theoretical contributions are:
1. We give a formal definition of desired equivariances for general architectures (via Neural DAG Automorphisms, see Section 2.2). We believe this is highly valuable for the study of metanetworks — equivariance for simple MLPs and simple CNNs were derived ad-hoc, whereas we formalized the desired equivariance for general feedforward networks. Future works could develop additional metanetwork architectures that achieve this desired equivariance.
2. We design a general class of equivariant architectures using a completely different approach than previous work. Our approach hinges on graph constructions rather than equivariant linear maps. We prove theoretically that our method gives the desired equivariance, a result that takes significant effort to produce for these prior works (Proposition 2).
3. Though our method can process many more types of input networks than previous work, we show that our method theoretically does not lose any expressive power on the simple input networks that previous works are restricted to (Propositions 3 and 4). Indeed, our networks are provably at least as expressive as StatNN and NP-NFN.

**Experiments**

Several reviewers suggested that we compare with more baseline methods, and generally include more experiments. We include two new sets of experiments in the revision:  (1) predicting accuracy of simple CNNs without normalization layers (a restricted data type that NFN [Zhou et al. 2023a] and NFT [Zhou et al. 2023b] can process), and (2) editing of 2D-INRs, which are simple MLPs [Zhou et al. 2023a]. For both, we compare against NFN and NFT. Also, (2) is an application that was not in our original submission.


(1) Predicting generalization performance on small CNNs trained on CIFAR-10-GrayScale (higher is better). Here we see that our GMN matches the performance of NFN-HNP, and outperforms all other methods.

| Method | Test $\tau$ |
| --- | --- |
| NFN-HNP | **.934$\pm.001$**  |
| NFN-NP | .922$\pm.001$ |
| StatNN | .915$\pm.002$ |
| NFT | .926$\pm.001$ |
| GMN (ours) | **.933$\pm.001$** |

(2) 2D INR Editing (lower is better). GMN beats all methods on contrast, and outperforms all methods besides NFT on dilation.

| Method | Contrast | Dilate |
| --- | --- | --- |
| MLP | .031 | .306 |
| MLP-Aug | .029 | .307 |
| NFN-PT | .029 | .197 |
| NFN-HNP | .0204 | .0706 |
| NFN - NP | .0203 | .0693 |
| NFT | .0200 | **.0510** |
| GMN (Ours) | **.0198** | .063 |


Overall, we see that our Graph Metanetworks beat most methods on these simple input data types. Once again, we stress that our method also generalizes to input neural architectures that are much more diverse than previous work, e.g. inputs with attention layers, normalization layers, and general group-equivariant linear maps. Further, any graph neural network advance can be used for our GMNs, so the performance of GMNs can surely be improved with more exploration of the GNN design space.

---

### Author Response · Authors · 2023-11-22

We sincerely appreciate the thoughtful feedback from the reviewers. We carefully accounted for all comments and have uploaded a revised version that addresses the major concerns raised. Notably, we added new experiments, as requested, that compare to new baselines and study new tasks --- further validating our approach.

If the reviewers feel that we have adequately addressed their concerns, we would be grateful if they would consider reassessing their rating.

We would appreciate the opportunity to clarify or expand any part of the updated manuscript while the discussion period remains open. Thank you.

---

### Meta-Review · Area_Chair_z4gN · 2023-12-10

**Metareview:**

This paper presents a method for learning from neural networks by taking their weights as inputs and representing them as graphs and using graph neural networks to process them. The main claimed contribution is that of handing a much more diverse set of neural network inputs. Evaluation is done by learning to predict the accuracy of models on various datasets.

The reviewers found this paper addresses an interesting and promising topic, describing as impressive, handling scaling issues and, more importantly, expanding on the variety of representable networks. The paper also presents some theory on the expressiveness and equivariance of the proposed approach and it seems the reviewers found the overall description of the method very informative. The main weaknesses of the paper seemed to be on requiring additional experiments, more comparisons, running ablation studies to understand how the model behaves under different settings and reproducibility. The authors have addressed some of these concerns and have promised to work on the ablations and making their code available in the future. I recommend this paper for acceptance without hesitation but strongly encourage the authors to include these latter 2 improvements by the time of the final submission.

**Justification For Why Not Higher Score:**

This paper seems very interesting in terms of the problem and also the methodology  buy I am a little concerned about the reviewers' confidence hence recommending only spotlight.

**Justification For Why Not Lower Score:**

Interesting enough to the wider ICLR community to be presented as a spotlight.

---

### Decision · Program_Chairs · 2024-01-16

Accept (spotlight)